

# Interannual Variability in the Tropical Atlantic from the Last Glacial Maximum into Future Climate Projections simulated by CMIP5/PMIP3.

Chris Brierley[1] and Ilana Wainer[2]

[1]Environmental Change Research Centre, Department of Geography, University College London, Gower St, London, WC1E 6BT, United Kingdom
[2]Departamento de Oceanografia Física, Química e Geológica, Instituto Oceanográfico da Universidade de São Paulo, Praça do Oceanográfico, 05508-120, São Paulo, Brasil

*Correspondence to:* Chris Brierley (c.brierley@ucl.ac.uk)

**Abstract.**

Tropical Atlantic Variability (TAV) plays an important role in driving year-to-year changes in rainfall over Africa and South America. In this study, its response to global climate change is investigated through a series of multi-model experiments. We explore the leading modes of TAV during the his-

torical, last glacial maximum, mid-Holocene and future simulations in the multi-model ensemble known as PMIP3/CMIP5. Despite their known sea surface temperature biases, most of the models are able to capture the Tropical Atlantic's two leading modes of SST-variability patterns - the Atlantic Meridional Mode (AMM) and the Atlantic zonal mode (also called the Atlantic Niño or ATL3). The ensemble suggests that AMM amplitude was less during the mid-Holocene and increased during

the last glacial maximum; but is equivocal about future changes. ATL3 appears stronger under both the last glacial maximum and future climate changes, with little consistent message about the mid-Holocene. The patterns and the regions under the influence of the two modes alters under climate change - in concert with changes in the mean climate state. Both modes demonstrate a coupling with the equatorial Pacific that depends on the climate period being considered - especially for the ATL3

mode of equatorial Pacific. In the future climate experiment, the equatorial mode weakens, the whole northern hemisphere warms up while the south Atlantic displays an hemisphere-wide weak oscillating pattern. For the LGM, the AMM projects onto a pattern that resembles the Pan-Atlantic Decadal Oscillation. No robust relationships between the amplitude of the zonal and meridional temperature gradients and their respective variability was found.



## 1 Introduction

### 1.1 Tropical Atlantic variability and its importance

Variability in the Tropical Atlantic Ocean occurs at different time scales ranging from seasonal, interannual to decadal and longer (Xie and Carton, 2004; Chang et al., 2006; Wainer et al., 2008; Deser et al., 2010; Muñoz et al., 2012). The dominant frequency for this region is the seasonal cycle, which combined with continental monsoon forcing and air-sea interaction, regulates the latitudinal displacement of the rain producing Inter-Tropical Convergence Zone (ITCZ). The marine portion of the ITCZ, in its turn, is locked to the Atlantic's sea surface temperature (SST). In the Tropical Atlantic, changes in SST are in phase with the meridional displacement of the ITCZ and associated meridional wind stress (Wainer and Soares, 1997). SST departures from the seasonal cycle are primarily driven by changes in surface winds that result from local air-sea interaction associated with the latitudinal migration of the ITCZ, or remotely forced by external factors (e.g. variability associated with ENSO). There is also significant interannual variability in the Tropical Atlantic that is represented by its two leading SST modes.

The first mode is a meridional mode characterized by a north-south inter-hemispheric SST gradient (Atlantic Meridional Mode, hereafter referred to as - AMM), which significantly impacts changes in the position and strength of the ITCZ. Studies have linked this mode to changes in the surface winds and associated evaporation feedbacks (Xie and Carton, 2004; Mahajan et al., 2010; Amaya et al., 2016). These feedbacks are important in defining the spatial and temporal features of the AMM and its impact on rainfall. It is associated with a shift in the distribution of precipitation towards the hemisphere with anomalously warm SST (relative to the other). The inter hemispheric SST-gradient is accompanied by a cross-equatorial atmospheric flow in the same direction (Chiang et al., 2002; Saravanan and Chang, 2004; Xie and Carton, 2004).

The second mode of Tropical Atlantic Variability is a zonal mode (ATL3; Zebiak, 1993) that is governed by equatorial ocean dynamics in response to surface winds; much like ENSO in the Pacific. It involves changes in the Atlantic equatorial cold tongue and associated displacement of the equatorial thermocline. Although this mode has a weaker impact on the meridional displacement of the ITCZ, it can nonetheless impact South America precipitation (Tokinaga and Xie, 2011). The positive phase of this equatorial mode presents positive SST anomalies in the eastern part of the basin and is associated with increased precipitation over Northeast Brazil and the western Amazon. The negative phase is associated with weakening of the African monsoon.

In both modes, the coupling between SST and the ITCZ is an important driver of rainfall variability both for North-Northeast Brazil and for the Sahel region in Africa. It is regulated by the combined changes in intensity and meridional displacement of the ITCZ driven by the underlying SST gradient associated with a surface wind response (Ruiz-Barradas et al., 2000; Servain et al., 2000; Okumura





and Xie, 2004). The reader is referred to Xie and Carton (2004) for a detailed review of the patterns,
mechanisms and impacts of Tropical Atlantic Variability.

### 1.2 What do we know about TAV in past climates?

Several studies of past climates have attempted to understand predominant characteristics of the Last
Glacial Maximum (LGM) and mid-Holocene relative to preindustrial controls (PI) (e.g. Pinot et al.
(1999); Wainer et al. (2005); Braconnot et al. (2007); McGee et al. (2014); Donohoe et al. (2013)
among others). A common thread among these studies is the idea that in response to the changes
in meridional SST-gradient, the mean position of the ITCZ shifts to the hemisphere with warmer
temperatures (c.f. the AMM) It is known that during the LGM the tropics cooled less than extra-
tropical latitudes (e.g. Pinot et al., 1999; Braconnot et al., 2007; Wainer et al., 2005). Braconnot et al.
(2007) examine and quantify changes in the north-south location of the ITCZ from simulation results
from PMIP2 for the Last Glacial Maximum (LGM) and the Mid-Holocene (MH). They establish that
changes in the associated meridional SST-gradient in the tropical Atlantic during summer at the MH
are in phase with changes in precipitation over West Africa. Wainer et al. (2005) discuss that for
the LGM, the marine portion of the ITCZ does not reach the South American continent during DJF
contributing to weakened precipitation. McGee et al. (2014) find that for the LGM and mid-Holocene
the latitudinal shift in the mean ITCZ is less than $1°$ latitude. They discuss how the position of the
ITCZ is associated with the heat transport between the hemispheres. An important conclusion of
their work (also noted by Donohoe et al., 2013) is that tropical SST gradients for past climates can
be reconstructed with greater certainty than the ITCZ position, which means that understanding the
fluctuations of the anomalous SST variability patterns allows the assessment of past changes in ITCZ
position and related rainfall patterns.

Considering the significant impact that Tropical Atlantic Variability (TAV) has on the the posi-
tioning of the ITCZ and the distribution of rainfall of the adjacent continental regions, and given that
it has been changing with global climate change, we seek to characterize the SST-modes of TAV for
different climates. The idea is to identify, if any, changes in TAV for past climates and understand
its behavior in future projections using simulations from complex climate models. The present study
has the goal to examine the performance of Earth System Models relative to the simulation of TAV
in terms of SST for different climates, in the context of the Palaeoclimate Model Intercomparison
Project (PMIP). Hopefully by understanding the link between the modes of variability of the Tropi-
cal Atlantic for different climates, we can improve our understanding of related monsoon-dynamics
and mechanisms in the region.





## 2 Methods

### 2.1 Model simulations

Coupled atmosphere-ocean general circulations models (GCMs) are routinely used for climate re-
search. Simulations of future climate are coordinated by the Coupled Model Intercomparison Project
(CMIP) through the use of collectively defined experiments (Taylor et al., 2011). The fifth phase of
CMIP was heavily relied upon by the IPCC for their fifth assessment report (IPCC, 2014). Addition-
ally, a series of past climate experiments have been coordinated by PMIP. Three such experiments
formed part of the $3^{rd}$ phase of PMIP: the mid-Holocene, the last glacial maximum and last mil-
lennium (although this latter one is not analysed here). A pre-industrial control and an idealized
warming scenario were also requested to establish the baseline and forced climate response respec-
tively. Further details of these experiments will be introduced later when relevant.

Anomalous SSTs are calculated separately for each individual simulation. The climatology across
the all years is used for the preindustrial, midHolocene and LGM experiment; the average of 1971-
2000 for the historical simulation; and the final 40 years of idealized warming experiment. For all
simulations, the resulting SST anomalies are then linearly detrended to remove any residual drift or
aliasing from changes in mean state.

Not every modeling group within PMIP was able to perform all the requested simulations - al-
though only in the case of GFDL's last glacial maximum run was this for scientific rather than
resourcing reasons. Here we investigate every simulation that has posted the required data on the
Earth System Grid Federation's data nodes (Table 1). Two modelling groups provided multiple re-
alisations of the simulations differing only by their initial conditions (Table 1). Every simulation is
considered equally likely during the creation of any ensemble averages. The ensemble-mean change
patterns shown consist of the average of the difference for each model that has run both simulations
(rather than the difference of the ensemble mean). The spread within in the ensemble is illustrated
throughout this analysis by stippling to indicate consistency. We consider the ensemble's signal to be
consistent if two-thirds or more of the participant models show a change of the same sign as the en-
semble mean. Whilst this particular measure is not overly stringent, it does allow ready identification
of regions where the signal is more likely to be robust.



| | Group | Model | piControl | historical | MH† | lgm | 1pctCO2 | past1000 |
|---|---|---|---|---|---|---|---|---|
| A | NCAR | CCSM4 | 1051 | 156 | 301 | 101 | 140 | 1001 |
| B | CNRM-CERFACS | CNRM-CM5 | 300 | 156 | 200 | 200 | 140 | - |
| C | FUB | COSMOS-ASO‡ | 399 | - | - | 599 | - | - |
| D | CSIRO-QCCCE | CSIRO-Mk3-6-0 | 500 | 156 | 100 | - | 140 | - |
| E | UNSW | CSIRO-Mk3L-1-2 | 1000 | - | 500 | - | 140 | 1000 |
| F | EC-EARTH-2-2 0 | - | - | 40 | - | - | | |
| G | LASG-CESS | FGOALS-g2 | 200 | 156 | 200 | 100 | 140 | - |
| H | LASG-CESS | FGOALS-s2 | 501 | - | 100 | - | 140 | 1001 |
| I | NASA GISS | GISS-E2-R* | 550 | 156 | 100 | 100 | 140 | - |
| J | NASA GISS | GISS-E2-R* | 505 | - | - | - | 140 | - |
| K | NASA GISS | GISS-E2-R* | 1163 | - | - | - | - | |
| L | NASA GISS | GISS-E2-R* | 100 | - | - | - | - | |
| M | NASA GISS | GISS-E2-R* | - | - | - | 100 | - | |
| N | NASA GISS | GISS-E2-R* | - | 156 | - | - | 140 | |
| O | UBris | HadCM3 | 1199 | - | - | - | - | |
| P | MOHC | HadGEM2-ES | 575 | 145 | 101 | - | 140 | - |
| Q | MOHC | HadGEM2-CC | 240 | 145 | - | - | - | |
| R | IPSL | IPSL-CM5A-LR | 1000 | 156 | 500 | 200 | 140 | 1001 |
| S | CAU-GEOMAR | KCM1-2-2‡ | 200 | - | 100 | - | 133 | - |
| T | MIROC | MIROC-ESM | 630 | 156 | 100 | 100 | 140 | 1000 |
| U | MPI-M | MPI-ESM-P* | 601 | 156 | 100 | 100 | 140 | 1000 |
| V | MPI-M | MPI-ESM-P* | 601 | - | 100 | 100 | - | - |
| W | MRI | MRI-CGCM3 | 350 | 156 | 100 | 100 | 140 | - |
| X | BCC | bcc-csm1-1 | 50 | 163 | 100 | - | 140 | 1001 |

**Table 1.** The number of simulated years of monthly output used to calculate the tropcial atlantic variability in each model simulation. The models can be identified by their established acronyms from the Earth System Grid Federation database. †The name of this simulation is 'midHolocene' on the Earth System Grid Federation. ‡ indicates models that only form part of PMIP3, but not CMIP5. *both GISS-E2-R and MPI-ESM-P deposited multiple ensemble members.



## 2.2 Observations

This research involves the joint investigation of sea surface temperature, surface air temperature and precipitation. We adopt a combination of the Twentieth Century Reanalysis (Compo et al., 2011) for the atmospheric variables with HadISST1.1 (Rayner et al., 2003) for the SST. HadISST1.1 forms the underlying boundary conidtions for the Twentieth Century Reanalysis (Compo et al., 2011), providing internal consistency between the datasets. These datasets exist over the period 1871-2012 C.E., although there is an increased amount of uncertainty in the early portion of the record (e.g. Ilyas et al., 2017). We follow Solomon et al. (2007) in using a climatological period of 1971-2000, as the historical simulations only extend until 2005.

## 2.3 Definition of modes

Climate modes of variability are preferred spatial patterns associated with time variations that have global-to-regional impacts. Both modes of tropical Atlantic variability analysed here have been identified using area-averaged SST anomaly indices. We avoid using definitions based upon Empirical Orthogonal Functions (EOFs) as preliminary analysis indicated they could have alternate ordering in the various models and simulations.

PMIP's Paleovariability working group is endeavoring to perform routine evaluation of the simulated climate variability (Kageyama et al., 2018) using the ESMValTool software (Eyring et al., 2016). This includes a collection of standardized analyses to look at coupled climate modes (Phillips et al., 2014). We have expanded the software to additionally incorporate analysis of the predominant modes of tropical Atlantic climate variability (TAV) listed below. The main source code is freely available at http://www2.cesm.ucar.edu/working-groups/cvcwg/cvdp. The full results of the software on the simulations described here are visible via the PMIP variability database at http://www2.geog.ucl.ac.uk/~ucfaccb/PMIPVarData/.

### 2.3.1 Atlantic meridional mode - AMM

The AMM is the leading mode of variability in the Atlantic. It represents variations in the north-south SST-gradient that exhibits opposite SST anomalies on either side of the mean position of the ITCZ (Servain et al., 1999). The underlying SST distribution has an influence on the position of the ITCZ which in turn affects the regional rainfall distribution. Here we adapt the SST-based index of Servain et al. (1999). The AMM state is defined as the basin-wide area-average, detrended SST anomaly difference between the two hemispheres. More precisely is the average of [20°N-5°N, 80°W-0°W] minus the average of [10°S-5°N, 60°W-15°E].



### 2.3.2 Atlantic zonal mode - ATL3

The Atlantic zonal mode (Zebiak, 1993) is the second mode of tropical Atlantic variability and represents changes in the cold tongue at the eastern part of the basin, just south of the equator. We adopt the ATL3 region of Zebiak (1993) as a metric. It is defined to be the area-average of the detrended SST anomaly over the region [3°N-3°S, 20°W-0°W]. This index definition is somewhat analogous to that of a Niño region in the Pacific, leading to this mode sometimes being term the 'Atlantic Niño' (Tokinaga and Xie, 2011; Xie and Carton, 2004).

## 3 Present-day

### 3.1 Mean State

Prior to investigating the interannual variability, it will be instructive to look at the representation of the mean climate state. The Atlantic is warmest on the Equator throughout the year (Fig. 1). The West Atlantic warm pool shifts to stay in the summer hemisphere. The Eastern Equatorial Atlantic has a tongue of cold upwelling that peaks in JJA. The warmest SSTs are associated with the strong precipitation of the ITCZ (Fig. 2). Both South America and West Africa experience heavy monsoonal rainfall in their respective summers.

General circulation models provide our best tool for modeling the climate and generally provide a fair representation. However, all models have some biases in their mean climate state. On the ensemble mean, models are unable to simulate the correct magnitude of equatorial upwelling (Fig. 1). Furthermore, the West Atlantic warm pool extent is underestimated. Both of these biases persist throughout the year. There are precipitation biases as well. In general, the models are unable to get the full intensity of the ITCZ, which is also associated with too much rain falling south of the ITCZ. The West African monsoon is biased dry, whilst the models simulate too much convection over N.E. Brazil. The ensemble mean biases discussed are relatively consistent within the ensemble (the majority of the biases demonstrated in Figs 1 and 2 are stippled - meaning that two-thirds or more of the models have the same sign bias as plotted). These biases have been reported by other studies that looked at Tropical Atlantic Variability (TAV) in CMIP models (Breugem et al., 2006; Siongco et al., 2015; Richter et al., 2014; Deppenmeier et al., 2016; Wang et al., 2017). Consistently, the bias is related, in many of the models to either a weak eastern equatorial cold tongue or failure to reproduce it. Breugem et al. (2006) examines 20th Century simulations in nine GCMs and identify strong interactions between the Atlantic zonal and the meridional modes that are not realistic. They discuss that the models that seem to best represent the meridional mode show its weakening in future climate conditions. Siongco et al. (2015) examined precipitation from 22 atmosphere-only models and identify an annual mean east-west bias associated with the ITCZ. They find that models with the East Atlantic bias tend to be high resolution models which rain excessively over the Gulf of



Guinea. Richter et al. (2014) analyzes the simulation results of 33 models of which 29 display biases relative to the mean state that include an annual mean zonal equatorial SST gradient whose sign is opposite to observations. Deppenmeier et al. (2016) compare the pre-industrial simulation results of 36 different models and show that although there are errors in the annual cycles of SST, wind-stress

and heat content, the relationship between them is well simulated. More recently, Wang et al. (2017) consider the validity of eastern equatorial Atlantic upwelling in the CMIP5 models when discussing their ability to predict the cold tongue SST development. Despite the mean state biases reported, the models are able to reproduce the dominant modes of climate variability of the Tropical Atlantic.



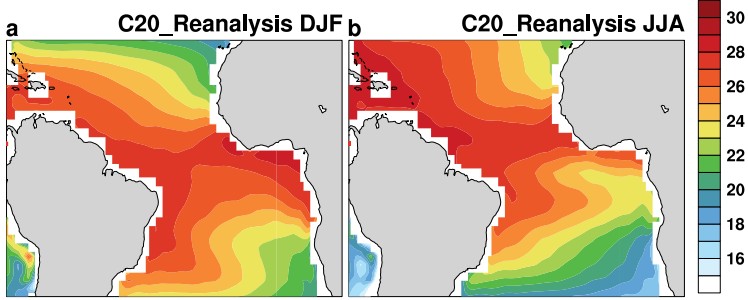

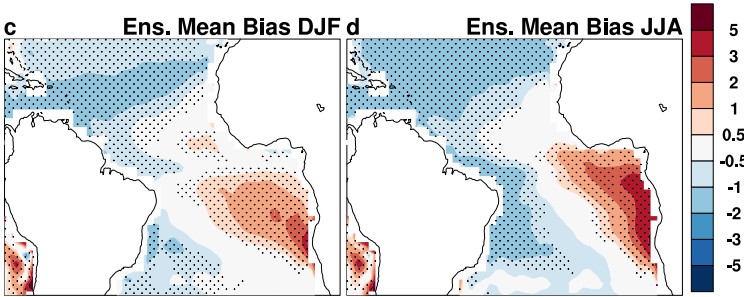

**Figure 1.** The sea surface temperature representation and changes in the ensemble. The climatological (1971-2000) sea surface temperature seen in HadISST (Rayner et al., 2003) in DJF (a) and JJA (b). Even on the ensemble mean, there are model biases in the seasonal temperatures in the historical simulation for DJF (c) and JJA (d). Stippling indicates regions where two-thirds or more of the models agree with the sign of the ensemble mean bias.



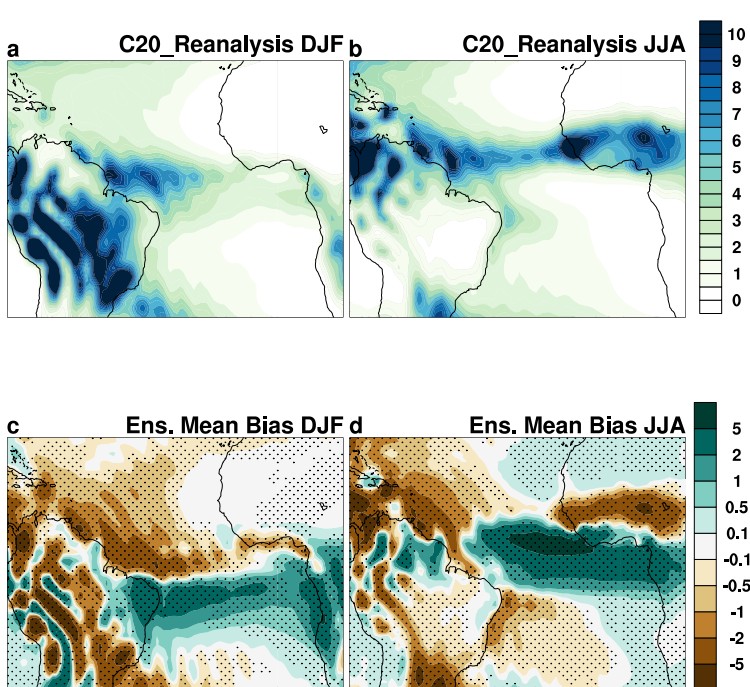

**Figure 2.** The seasonal cycle in precipitation and its changes across the ensemble. The climatological (1971-2000) precipitation seen in the Twentieth Century Reanalysis (Compo et al., 2011) in DJF (a) and JJA (b). The ensemble mean biases in the seasonal precipitation in the historical simulation for DJF (c) and JJA (d). Stippling indicates regions where two-thirds or more of the models agree with the sign of the ensemble mean bias.



### 3.2 Representation of Tropical Atlantic Variability

To address the question of how AMM and ATL3 are simulated, we present a comparison between the
ensemble mean pattern seen in the historical simulations (Table 1) and the HadISST temperature ob-
servations (Fig. 3). The time-series for both modes are determined through area-averaged, detrended
SST anomalies for both the AMM (sect. 2.3.1) and ATL3 (sect. 2.3.2). The standard deviation of
the resulting monthly time-series are calculated and shown in each panel (Fig. 3). The amplitude

variations of the ATL3 region are 0.18 °C, which is approximately identical to the ensemble mean
amplitude of 0.17 °C (given the ensemble spread). The standard deviation of the anomalous SST
gradient used as a metric for the AMM is substantially weaker than the variations of the ATL3 box
alone. Nonetheless the observed amplitude of 0.05 °C compares favorably the ensemble mean value
of 0.04 °C. It should be noted that one should expect the GCMs to sample different phases of the

low frequency natural variability, so a direct comparison of the time-series is not appropriate. Addi-
tionally, there are uncertainties in the observational record, which may be considerable in the early
portion of the record (Rayner et al., 2003; Compo et al., 2011; Ilyas et al., 2017).

The spatial patterns associated with the tropical Atlantic variability are demonstrated through
simple linear regression of the area-averaged indices onto the monthly sea surface temperatures (Fig.

3). This regression is extended across the globe, which highlights correlations with other modes of
internal variability. This does not imply that a causal relationship extending out of the Atlantic to
other ocean basins exists.

The SST pattern associated with the AMM in the GCM ensemble appears to be well represented
in the Atlantic when compared to the HadISST data set (Fig. 3). There is some smoothing of the

dipole structure in the South Atlantic, which is perhaps to be expected given the ensemble-mean
biases in the mean state of the region (Fig. 1). The most visually striking difference between the
ensemble mean AMM pattern and the observations occurs over the Pacific. Here the observations
show a structure reminiscent of the Interdecadal Pacific Oscillation (Power et al., 1999), which leads
to regression strengths as strong in the Central Pacific as in the Atlantic. It is unclear if this arises

from a missing connection in the GCMs or a deficient sampling of low frequency variability in the
observations. Nonetheless the ensemble clearly demonstrates some coupling between the equatorial
Pacific (ENSO) and the AMM (Saravanan and Chang, 2000; Keenlyside and Latif, 2007; Rodríguez-
Fonseca et al., 2009).

The spatial extent of the ATL3 does not extend far beyond the tropical Atlantic (Fig. 3). In fact, in

both observations and models it has little effect on the North Atlantic. The projection of the ATL3 in
models is predominantly onto the Equator itself and there is a muted effect on the upwelling region.
This is likely due to an under-representation of the upwelling in the model as demonstrated by the
substantial warm biases in the mean state (Fig. 1).





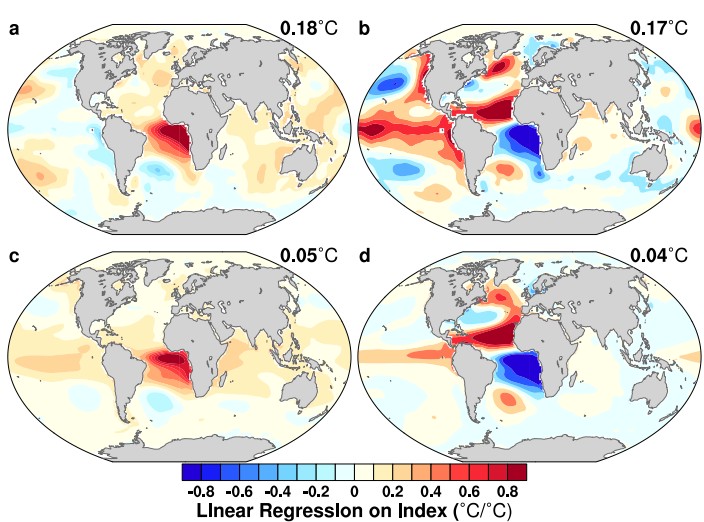

**Figure 3.** Sea surface temperature patterns related to ATL3 (a,b) and AMM (c,d) indices in observations (left panels) and CMIP5/PMIP3 (right). The model panels show the ensemble mean patterns for the the historical simulations. The standard deviations of the SST indices are also shown.

## 4   Past Climates

### 4.1   Mid-Holocene


Around 6,000 years ago was the warmest portion of the Holocene (Marcott et al., 2013) - although there are suggestions this may only represent the summer rather than annual average temperatures (Liu et al., 2014). The magnitude of the simulated temperature changes relative to preindustrial





conditions were comparatively small (Fig. 4a, b), with several areas of cooling on the Equator. The

climate change was caused by differences in the orbital precession that drove movement of the

ITCZ's seasonal cycle to favour the Northern Hemisphere (Braconnot et al., 2007). Most notably,

this increased the precipitation over Northern Africa and supported green vegetation in the Sahara

(Hély et al., 2014). The ensemble simulates a noticeable northward shift in precipitation over Africa

(Fig. 5b). This is however significantly less than observed in the region for the mid-Holocene (Perez-

Sanz et al., 2014). It has been shown that a more accurate simulation of mid-Holocene vegetation

changes in the Sahara can influence the interannual climate variability (Pausata et al., 2017). Over

N.E. Brazil, the monsoon rainfall reductions are relatively moderate (Fig. 5b), although there is a

general decrease in summer rainfall across South America (Fig. 5a).





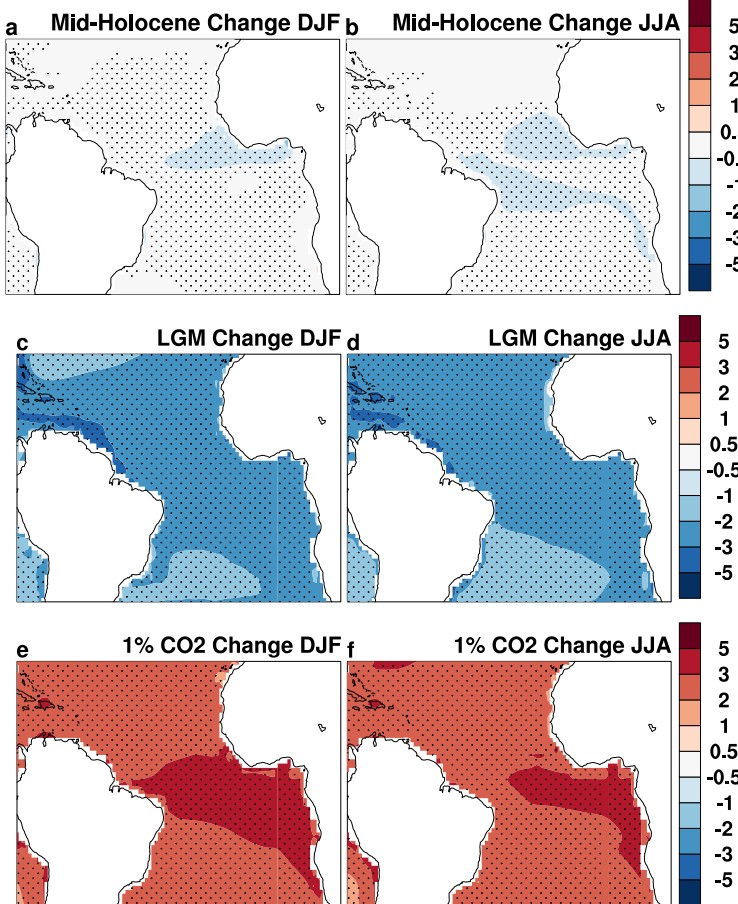

**Figure 4.** The sea surface temperature changes shown by the ensemble. The ensemble mean difference between the mid-Holocene and preindustrial simulations demonstrates the temperature impacts in DJF (a) and JJA (b). The last glacial maximum is simulated as being substantially colder than the preindustrial in both DJF (c) and JJA (d). In contrast, the ensemble mean average of the final forty years of the 1% per year increasing carbon dioxide concentration run is warmer than preindustrial in both DJF (e) and JJA (f). Stippling indicates regions where two-thirds or more of the models agree with the sign of the ensemble mean change.

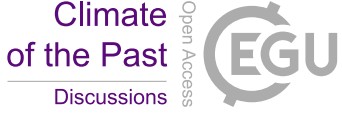

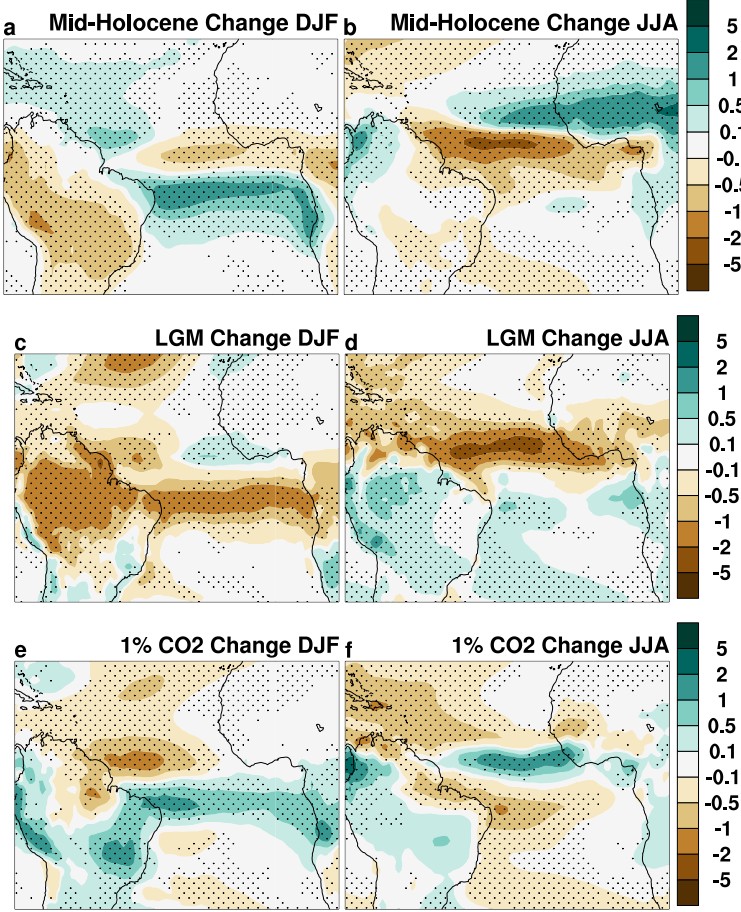

**Figure 5.** The changes in seasonal cycle in precipitation across the ensemble. The ensemble mean difference between the mid-Holocene and preindustrial simulations shows the movement of the ITCZ in DJF (a) and JJA (b). The last glacial maximum is simulated as having less intense rain bands than the preindustrial in both DJF (g) and JJA (h). In contrast, the ensemble mean average of the final forty years of the 1% per year increasing carbon dioxide concentration run demonstrates enhanced activity over the ITCZ than in preindustrial for both DJF (i) and JJA (j). Stippling indicates regions where two-thirds or more of the models agree with the sign of the ensemble mean change.





The precession-related changes in the mid-Holocene led to changes in the amplitude of Tropical
Atlantic variability in many of the ensemble members (Fig. 6). These changes rarely exceed a 20%
reduction or increase in amplitude. More than two-thirds of the simulations show a reduction in
amplitude of the AMM (Fig. 6c), with a mean reduction of 8.6%. The ensemble is less confident
about the response of the ATL3 during the mid-Holocene. The ensemble mean change of 0.25% is
heavily influenced by the dramatic changes seen in KCM1-2-2 (Fig. 6).

The local spatial patterns associated with Tropical Atlantic Variability shift at the mid-Holocene
(Fig. 7), although the remote relationships appear visually more striking. A mid-Holocene weaken-
ing of the El Niño-Southern Oscillation has been seen in observations and models (Clement et al.,
2000; Chiang, 2009; Cobb et al., 2013). Despite this, there is a greater relationship seen in the ensem-
ble between the AMM and this region (Fig. 7b). This leads to an AMM pattern in the mid-Holocene
that is a better fit to the historical observations (Fig. 3). The mid-Holocene AMM sees a poleward
shift in its pattern over the North Atlantic, which is likely related to the shift in the mean-state of
the ITCZ (Fig. 5a, b). The ATL3 similarly shows signs of a Northward shift in its spatial pattern
(Fig. 7a). The mid-Holocene is simulated as having an increasingly inverse relationship between the
Atlantic and Pacific Niños.





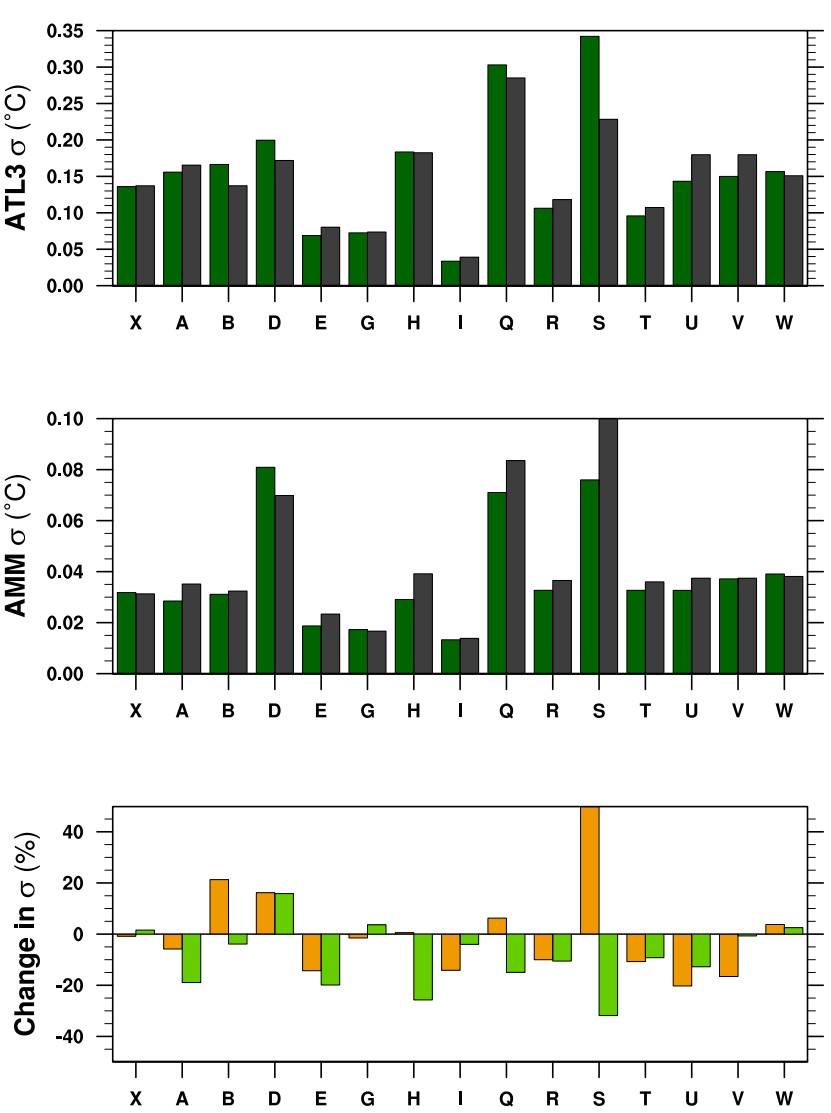

**Figure 6.** The standard deviations of the (a) ATL3 and (b) AMM indices in the mid-Holocene (green) and preindustrial results (grey); (c) changes expressed as a percentage across the ensemble for both ATL3 (yellow) and AMM (light green).





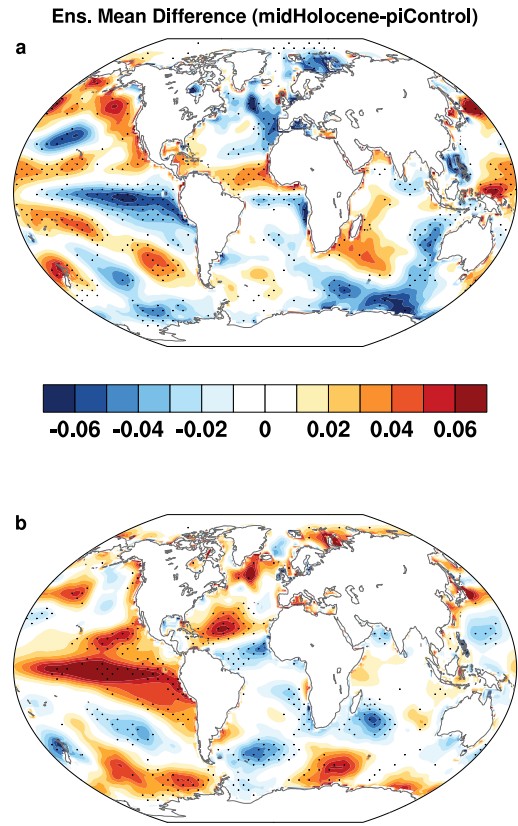

**Figure 7.** Mid-Holocene changes relative to preindustrial in the temperature patterns related to (a) the ATL3, (b) AMM indices. The changes are stippled with greater than two-thirds of the ensemble shows the same direction as the ensemble mean change. The contours show the change in regression strength onto the index in question; i.e. the change in local expression of +1°C index



## 4.2 Last Glacial Maximum

21,000 years ago saw the maximum extent of the ice-sheets during the last glacial. The orbital configuration then differed only slightly from the preindustrial. The large ice-sheets were accompanied by substantial cooling across the globe (Broccoli and Manabe, 1987; Clark et al., 2009; Annan and Hargreaves, 2013). Tropical sea surface temperatures cooled by roughly 2 °C (Fig. 4), predominantly controlled by the roughly 100 ppm drop in $CO_2$ (Broccoli, 2000; Marcott et al., 2014; Annan and Hargreaves, 2015).

The patterns of SST change are approximately uniform, although there is a slight weakening of the north-south gradient in the Tropical Atlantic (Fig. 4). The ensemble is equivocal about changes in the equatorial zonal SST gradient. The intensity of the tropical rainfall was generally reduced and the position of the ITCZ moved marginally southward (Fig. 5). The ensemble shows a strong propensity for increased climate variability in the Tropical Atlantic (Fig. 8). All but two models show an increasing amplitude of the AMM, with the ensemble mean increase being 22.4%. There is even more confidence in the increasing amplitude of the ATL3 (Fig. 8) - with only one dissenter. The average increase in the ATL3 amplitude is 32.8%.





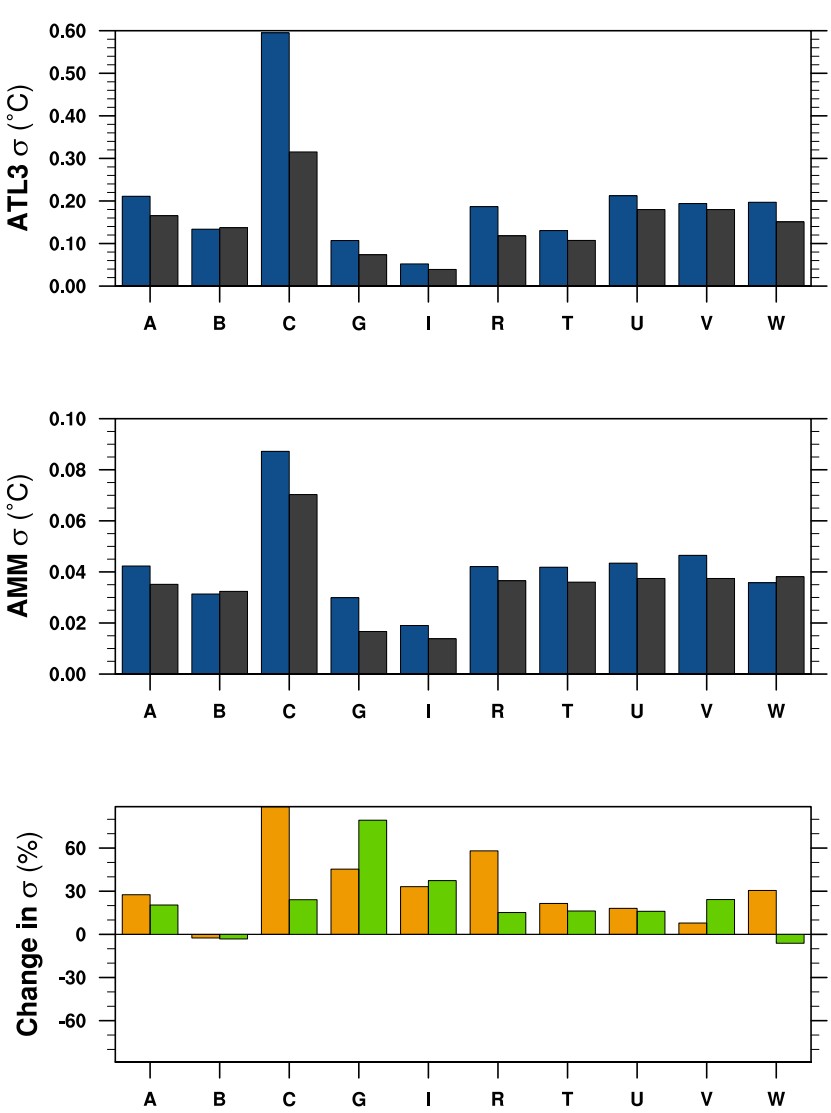

**Figure 8.** The standard deviations of (a) the ATL3 and (b) AMM indices in the last glacial maximum simulations (blue), shown alongside the values from their respective preindustrial control simulation (grey); (c) changes expressed as a percentage across the ensemble for both ATL3 (yellow) and AMM (light green).



These increases in amplitude are associated with robust changes in the spatial pattern of the modes
        (Fig. 9). The AMM sees an increasing influence over the South Atlantic (Fig. 9b). The North Atlantic
        has something similar, but likely overlaid with changes caused by the imposition of large ice-sheets
        over North America impacting the atmospheric dynamics (Pausata et al., 2011). The last glacial
        maximum sees a slight reduction in influence of the ATL3 in the Equatorial Atlantic (Fig. 9). We
interpret that to represent the ATL3 further constricting onto the Equator as the ITCZ moves slightly
        southward (Fig. 5). Further afield, ATL3 loses what connection it had to the equatorial Pacific (Fig.
        9).



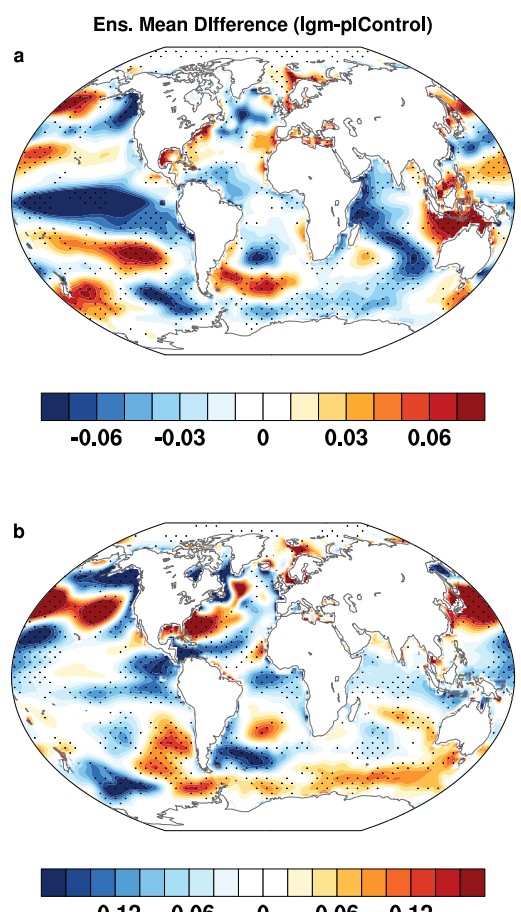

**Figure 9.** Last Glacial Maximum changes in the temperature patterns related to (a) the ATL3 and (b) AMM indices. The changes are stippled where greater than two-thirds of the ensemble showing the same direction as the ensemble mean change. The contours show the change in regression strength onto the index in question; i.e. the change in local expression of +1°C index

### 4.3 Future changes

The climate simulated for both mid-Holocene (sect. 4.1) and last glacial maximum (sect. 4.2) represent equilibrated conditions between the climate and its forcing. The climate is expected to be still be a transient state throughout the coming century. Rather than selecting a particular plausible future scenario, we analyze the idealised simulations where the atmospheric $CO_2$ concentrations are





increased by 1% per year until it is quadrupled (Taylor et al., 2011). The mean climate during the final forty years of these simulations is substantially warmer (Fig. 4) with an intensified hydrologi-

cal cycle (Fig. 5). To have sufficient years to assemble robust SST patterns of climate variability, we consider the full length of these transient simulations having first removed a linear trend from each model grid point (after Phillips et al., 2014).

The mean SST and rainfall patterns are very similar to an opposite of those for the cold LGM (Fig. 4,5). However, the changes in Tropical Atlantic variability are not. There is an indication that there

will be an increase in amplitude of ATL3 (Fig. 10a), with an average increase of 12.0%. However, the ensemble is split evenly as to whether the AMM (Fig. 10b) will increase in amplitude (mean change of +2.8%). Despite that, there is a robust poleward expansion of the AMM's influence in the Atlantic (Fig. 11b). The influence of ATL3 also expands polewards, but only in the Northern Hemisphere (Fig 11a). There is a weakening of the connection between the AMM and Pacific El

Niño, although the ATL3 becomes more tightly connected (Fig. 11a). The relationship between the equatorial Atlantic cold-tongue and the tropical Pacific is relatively understudied and a consensus has yet to be reached upon it. Kucharski et al. (2011) demonstrate that the Pacific ocean response to Atlantic warming is a La Niña-like cooling response, much like the AMM-related future changes in Fig. 11b.





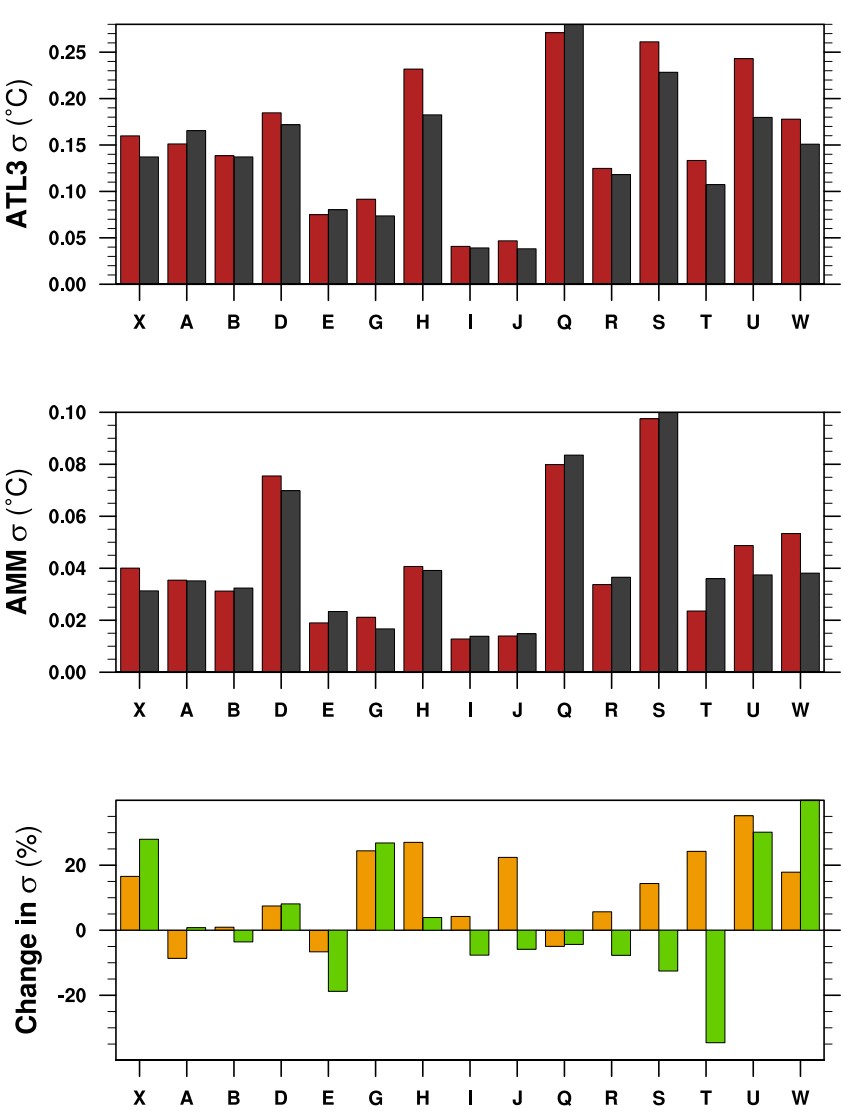

**Figure 10.** The standard deviations of (a) the ATL3 and (b) AMM indices in the 1% per year until quadrupled $CO_2$ experiment (red), shown alongside the values from their respective preindustrial control simulation (grey); (c) changes expressed as a percentage across the ensemble for both ATL3 (yellow) and AMM (light green).





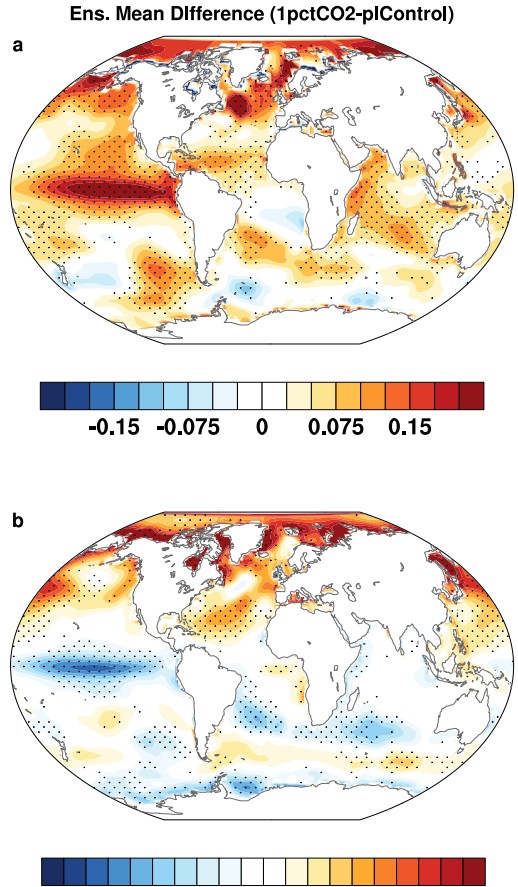

**Figure 11.** 1% per year until quadrupled $CO_2$ forced changes in the temperature patterns related to (a) the ATL3 and (b) AMM indices. The changes are stippled where greater than two-thirds of the ensemble shows the same direction as the ensemble mean change. The contours show the change in regression strength onto the index in question; i.e. the change in local expression of +1°C index



## 5 Discussion

The study of past climate change to place future changes in context is itself worthwhile (Pancost, 2017). The advantage of using models is that they can be equally applied to the past and future (as shown here). Ideally one could use observations of past climates to constrain future projections (Hargreaves et al., 2012). Unfortunately there are currently no reconstructions of past Tropical Atlantic variability to form that constraint. An alternate approach would be to detect an emergent relationship between the mean state and the climate variability. Reconstructions of changes in the mean state could then be used as emergent constraints on the future behaviour (Hall and Qu, 2006).

Changes in the amplitude of the ATL3 mode have previously been linked to changes in the zonal SST gradient since 1950 (Tokinaga and Xie, 2011). In Fig. 12, we investigate whether this link holds across the ensemble and multiple climates. We use the difference in the area-averaged SST between [3°N-3°S, 45°W-25°W] and [3°N-3°S, 20°W-0°W] to characterize the west-east SST gradient (after Tokinaga and Xie, 2011), and only consider the climate change signal to prevent the model biases hiding any relationship. The future simulations are only analyzed over the final forty years - once the main climate change signal is dominant (unlike in sect. 4.3).

Despite the ensemble showing robust changes in mean state and often robust changes in variability, there is little relationship emerging between the two pairs of changes (Fig. 12). This perhaps questions the previous conclusions of Tokinaga and Xie (2011). In fact, extending their time series of ATL3 earlier within the instrumental period indicates little persistence of the trends they find between 1950-2000 (not shown). However, Tokinaga and Xie (2011) highlight aerosols as the cause of their trends. Yet aerosols and their associated forcing are not properly explored across these simulations, leading us to conclude that further work is required to understand the future of the ATL3.

The AMM is defined as variations in the inter-hemispheric tropical SST gradient (sect. 2.3.1). It would seem logical to think that as that inter-hemispheric gradient changes the interannual variability would also change. We explore this suggestion to search for an emergent constraint of the future AMM response. When looking at changes in the standard deviation of the AMM amplitude as a function of the meridional gradient, there seems to be a disconnect from the mean state gradient (Fig. 13). A decrease in the AMM amplitude change can be associated with an increase in the meridional gradient for the mid-Holocene while the opposite occurs for the LGM. The 1% per year until quadrupled $CO_2$ experiment shows that the amplitude of the meridional mode decreases with decreasing gradient. Therefore there is little robust relationship overall.





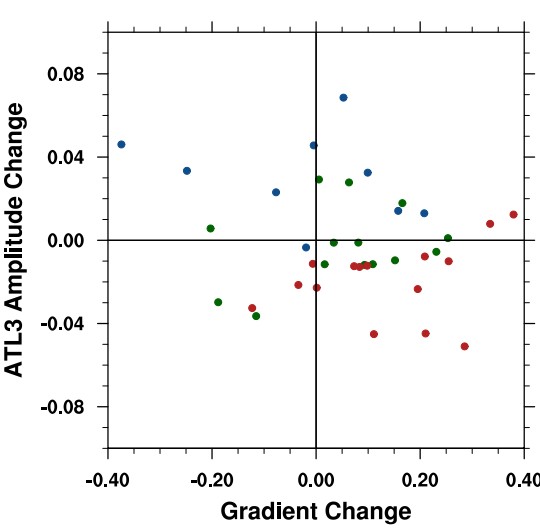

**Figure 12.** The change in standard deviation of the zonal mode (ALT3) as a function of the change in west-east SST gradient. The gradient is calculated using the Tokinaga and Xie (2011) regions (see text). The colors indicate the different experiments: 1pctCO2 (red), mid-Holocene (green) and last glacial maximum (blue).





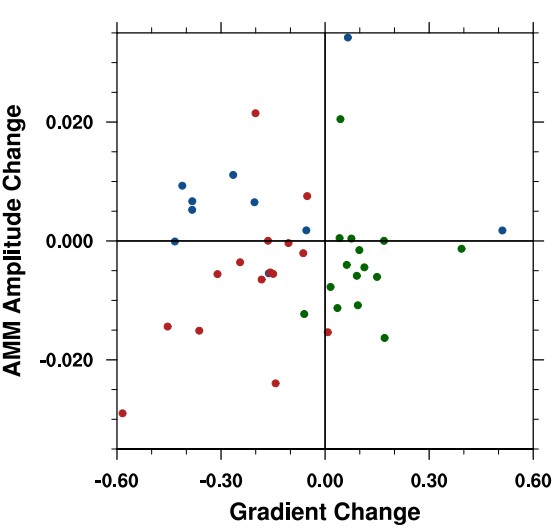

**Figure 13.** The change in standard deviation of the meridional mode (AMM) as a function of the change in meridional SST gradient. The gradient is calculated using the same regions as for the AMM index itself (sect. 2.3.1). The colors indicate the different experiments: 1pctCO2(red), mid-Holocene (green) and last glacial maximum (blue).



## 6   Conclusions

All models are able to represent the main characteristics of the dominant modes of variability, with
similar mean state bias, for all climate periods analyzed. These biases are consistent among all mod-
els (Fig. 1) especially in the equatorial cold tongue. They are also consistent in showing precipitation
biases over South America and Africa with associated biases in the ITCZ location. Despite their
mean state biases, the simulation results show reasonable representation of the observed patterns of
Tropical Atlantic variability. Overall, the ensemble analysis shows that SST anomalies patterns asso-
ciated with cooling north of the equator can be associated with a southward shift of the Intertropical
Convergence Zone (ITCZ) which is accompanied by a weakening of rainfall and increasing dryness
over both continents.

Across the climate periods, the relationship between equatorial Atlantic and Pacific variability is
not robust. The ensemble shows that this coupling changes substantially for all climate periods: For
the LGM, the ATL3-related equatorial cold tongue intensifies near the equator and weakens off the
coast of Namibia while the Pacific equatorial response significantly strengthens. The mid-Holocene
shows a stronger intensification of the cold tongue system. It also displays the increase of the Pacific
equatorial mode, but its less than for the LGM. The warm climate of the 1% per year until quadrupled
$CO_2$ experiment, shows opposite results: the equatorial Atlantic cold tongue weakens and there is
an associated increase of the Pacific equatorial mode. The region along the coast of South Africa
and Namibia shows cooling. In fact, Lübbecke and McPhaden (2012) discuss how the relationship
between SST anomalies in the equatorial Atlantic and Pacific may vary with time. ENSO events can
be associated with both warm or cold SST anomalies in the equatorial Atlantic cold tongue region.
More recently, Li et al. (2016) show, the importance of Atlantic SST anomalies for the tropical
Pacific over the past three decades.

Regarding the Meridional Mode (AMM), we also find a coupling with the equatorial Pacific for
all periods. The AMM for the LGM (Fig. 9b) projects onto a basin-wide meridional oscillating
pattern from Antarctica to Greenland with alternating signs, much like the Pan-Atlantic Decadal
Oscillation pattern (PADO) described by Xie and Tanimoto (1998). The AMM coupling with the
equatorial Pacific for the mid-holocene is strong and shows intensification. In the Atlantic, there are
small changes in the Tropics. A warming of the North Atlantic subtropics and cooling of the South
Atlantic except at the Brazil-Malvinas confluence region in the Southwest Atlantic. In the future
climate experiment, the equatorial mode weakens, the whole northern hemisphere warms up while
the south Atlantic displays an hemisphere-wide weak oscillating pattern.

This study has used the multi-model CMIP5/PMIP3 ensemble to investigate changes in Tropical
Atlantic variability across several climate states. Analysis of the idealised warming scenarios alone
suggest a spectrum of future climate change responses. The additional analysis of the palaeoclimate
simulations provides some valuable context for those responses. For example, the simulated future
ATL3 (Altantic Niño) amplitude increase is not simply a response to the warmer temperatures - as





a similar increase is seen during the last glacial maximum. The particular relationships between the mean state and variability changes that were tested here did not transpire to be robust. Nonetheless,

we feel the approach of analysing several different multi-model climate experiments, some with direct or proxy observations available, promises to constrain the uncertainty in future projections.

*Acknowledgements.* This analysis would not have been possible without the sterling effort by John Fasullo and Adam Phillips. Their foresight and generosity in building and freely distributing the Climate Variability Diagnostics Package is wonderfully refreshing. We acknowledge the World Climate Research Programme's Working

Group on Coupled Modelling, which is responsible for CMIP, and we thank the climate modeling groups (listed in Table 1 of this paper) for producing and making available their model output. For CMIP, the U.S. Department of Energy's Program for Climate Model Diagnosis and Intercomparison provides coordinating support and led development of software infrastructure in partnership with the Global Organization for Earth System Science Portals. This study was supported by the Belmont Forum's PACMEDY project through awards by NERC

(NE/P006752/1) and FAPESP (15/50686-1); I.W. was additionally supported by grants CNPq-301726/2013-2 and CNPq-405869/2013-4. All software is freely available at http://www2.cesm.ucar.edu/working-groups/cvcwg/cvdp and https://bitbucket.org/cbrierley/cvdp-synda. The results for individual models are freely available for inspection and download from the PMIP variability database at http://www2.geog.ucl.ac.uk/~ucfaccb/PMIPVarData/, along with results for many other modes of climate variability.





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
