# Peer review of "Interannual Variability in the Tropical Atlantic from the Last Glacial Maximum into Future Climate Projections simulated by CMIP5/PMIP3."

_Climate of the Past, 2017_

## Referee Comment (RC1) · Anonymous Referee #1 · 15 Jan 2018

This work investigated the change in the leading modes of the Tropical Atlantic Variability, the Atlantic Meridional Mode (AMM) and the Atlantico Niño (ATL3), in different climate scenarios: the historical, the last glacial maximum, the mid-holocene and future simulations in the multi-model ensemble of the PMIP3/CMIP5. Authors used this set of experiments in order to find robust signal of change in the Tropical Atlantic Variability. They found that all models across all experiments are able to represent main characteristics of dominant modes of variability in the Tropical Atlantic in spite of the mean state bias.

[Figure]

The paper addressed a relevant question: how the Tropical Atlantic Variability change under different climates and how the information from mean state and past climates can be used as a constraint for the future. They quantified first the mean state model bias of the tropical temperature and precipitation in the historical simulations against reanalyses. After that, they compared the magnitude of the simulated change of the tropical temperature and precipitation in the mid-holocene, the lgm and the future climate with the mean state bias, concluding that the simulated changes are reasonably respresented in these experiments. Hence, the main conclusion is that ATL3 and AMM are well represented among models and experiments considered, although authors found weak correlation with change in temperature gradients, so it is not possible to identify emerging constraints for future projections from this analysis.

I think this is a good work but sometimes the storyline is hard to follow: I suggest to work more on the structure of the paper and on discussion and conclusions in order to clarify main findings (perhaps merging both sections would be helpful).

About the method, I think that conclusions hold only if same models among experiments are considered, otherwise results might be affected by different model physics and also by the different number of the model used for each experiment (see my comments below).

Furthermore, several typos in the captions (see specific comments below) made the paper difficult to read.

Some recent and important literature is also missing (see the list below).

Minor Comments:

There are some typos in the text. Here, it is a list:

Ln 39: "It is associated with a shift. . .". It is not clear what is the subject of the sentence.

Ln 63: a full stop is missing after the brackets "(c.f. the AMM)".

Figure 1: the caption mentioned HadISST for panels a and b, but 20C_reanalysis is written on the top of both panels.

Figure 2: replace "in precipitation" with "of the precipitation" in the caption.

Ln 203: You specified already the acronym TAV for Tropical Atlantic Variability in the very beginning of the paper. You can use it throughout the paper.

Figure 3: Typos in the caption. ATL3 is shown in panel a and c. AMM in panel b and d. In the upper panels observation results and in the lower panels simulation results are shown. Also the standard deviations must be reorder: 0.17 refers to panel a, 0.18 to panel b, 0.05 to panel b and 0.04 to panel d.

Figure 6, 8, 10: It is better to change the color of dark gray bars in light gray bars. You could also replace the letters associated to each model, with a number or a short name.

Ln 280: repetition: " . . .is expected to be still be. . ."

Major Comments:

Ln 36: ". . .strength of the ITCZ". What do you mean for strength of the ITCZ. Usually, the ITCZ is the latitude of the wind convergence, and or, the latitude of the maximum of the precipitation. Please clarify.

Ln 40-42: Add literature about the ITCZ. (e.g. Schneider et al., 2014, Bischoff, T., & Schneider, T. , 2016, Green, B., & Marshall, J., 2017).

Ln 63: cite Schneider et al., 2014.

Ln 68: You might want to cite also D'Agostino et al., 2017. They linked Hadley Circulation changes also to change of the meridional temperature gradient and inter-hemispheric thermal contrast.

Ln 166-168: What do you mean for: "models are unable to get the full intensity of

the ITCZ"? I would like to clarify again that the ITCZ is a latitude of the maximum of the tropical precipitation not the rainfall intensity itself. Please reword, otherwise add specific analysis on the ITCZ shift, including also how to define the ITCZ in the methods.

Ln 225, the Mid-Holocene section: the Tropical Atlantic is elsewhere cooler than Pre-Industrial and this is pretty consistent among models. I was wondering if you believe to this result if the magnitude of the change is weaker than the change in the mean state.

Figure 4: I have a question about these panels. How did you perform the difference between the multimodel ensemble mean for each experiment and the multimodel ensemble mean of the Pre-industrial condition? You must use different pre-industrial multimodel ensemble mean for each experiments, because you must account for different model list. I did not find any specification about it in the paper. Furthermore, I have some doubt: I don't think the mean state bias can be used to give credibility to results of the different experiments in this contest, because the ensemble mean of the historical experiment accounts for 14 models. The ensemble mean of the pre-industrial for 21. They tell different story then. The difference is much more evident if you compute different pre-industrial ensemble mean for each experiment! Therefore, when you compare the climate change with the mean state bias, you are wrong because the different model list. I suggest to restrict the model list to common models only for all experiments. Unfortunately, the list is very short (only 9 models) but I think that is still possible to reach robust conclusions.

Ln 235: add also dust . . . cite Erger et al., 2016.

Ln 316: please quantify a "little relationship".

Ln 330: please quantify again "little robust relationship".

Ln 335-340: these statements about the ITCZ are not supported by the analysis shown in the paper. How did you quantify the ITCZ shift? Please include further analysis.

[Figure]

Suggested literature:

Bischoff, T., & Schneider, T. (2016). The equatorial energy balance, ITCZ position, and double-ITCZ bifurcations. Journal of Climate..

Green, B., & Marshall, J. (2017). Coupling of Trade Winds with Ocean Circulation Damps ITCZ Shifts. Journal of Climate.

D'Agostino, R., P. Lionello, O. Adam, and T. Schneider (2017), Factors controlling Hadley circulation changes from the Last Glacial Maximum to the end of the 21st century. Geophys. Res. Lett.

Egerer, S., Claussen, M., Reick, C., & Stanelle, T. (2016). The link between marine sediment records and changes in Holocene Saharan landscape: Simulating the dust cycle. Climate of the Past.

Schneider, T., Bischoff, T., & Haug, G. H. (2014). Migrations and dynamics of the intertropical convergence zone. Nature.

---

## Short Comment (SC1) · 7 Mar 2018

Dear Reviewer One,

Thank you for undertaking an assessment of our manuscript. We will reply to the points that you raise in detail separately. But I just wanted to clarify the method we have used to create the ensemble average changes.

We did not compute a preindustrial ensemble mean and a LGM ensmeble mean and then look at the difference between them. Rather we computed the change in each

model, interpolated these to a common grid, and then averaged across the changes.

We had felt this approach removed the requirement of having the same constituent models in each experiment's ensemble [after seeing IPCC projections with the number of models written on them, e.g. http://ar5-syr.ipcc.ch/topic$_{futurechanges.php}$

1. Be much more explicit about the ensemble averaging approach 2. Discuss the impact of just using the 9 common models on the results 3. Include the number of models used to make each figure in its (improved) caption 4. Provide a supplement of alternate figures using the other approach

Yours, Chris

---

## Referee Comment (RC2) · P. Braconnot (Referee) · 28 Mar 2018

Dear Author,

The reviews process of your manuscript it taking a long time, which is due to recurrent difficulties to get a second review. I am very sorry about it.

Since time is running I propose that you go for a major revision of your manuscript, taking into account the important comments of reviewer#1 who raised important issues on the clarity and organization of the paper, including methodological questions. I also

include below my own expertise of the manuscript, focusing on major aspects. If you decide to provide a revised version of the manuscript it will be sent for a second round of reviews, with the hope that we do not have to face a long delay similar to the one of the first round.

Best regards

Pascale Braconnot

Comments on the manuscript.

The subject is quite ambitious and timely, and the methodology used to discuss the two major Atlantic modes seems appropriate. As far as I know this has not been done yet, and providing systematic diagnoses to assess how the modes of variability are affected by climate change is a valuable task. In its present form however the paper is too descriptive and key aspects on precipitation are lacking. In particular :

- The introduction and first section highlight the fact that AMM and Atl3 modes have fundamental impact on South American and African monsoon, but this linkage is not discussed any further when considering the different climates. This limits the interest of the manuscript and is a major concern.

- The discussion on physical and dynamical mechanisms should be enlarged. This concerns both the anomalous circulations associate with the SST modes and the changes in these circulations associated with changes in mode patterns in the different climates

- One of the difficulties with the analyses of paleoclimate simulations is that both the background climate mean state and the variability change. How the pattern of the changes in variability is connected with patterns of the changes in the mean state should be discussed in more depth. A question out of this is does mode patterns only follow the mean state patterns? In other words if there is shift in the mode pattern is it directly reflecting a shift in the mean state pattern or is there other feedback that could explain that new areas become affected by the mode?

- The outline of the paper is also a little bit "boring". This feeling is due to the fact that the discussion section could include additional analyses to explain when possible part of the rationale behind model responses (which may be different from one period to the other). The discussion section could thus be enlarged and have a more appealing title and content. It could compare relationships as it is done as well as mechanisms. A few questions when reading the manuscript: it is interesting to see that the AMM mode is reduced at mid Holocene. Is it because the seasonal cycle is stronger and that a dipole-like pattern emerges in summer when comparing mid-Holocene with PI? Is there a reason why a colder climate would have increased variability? Could the non-symmetrical differences between LGM and future results from non-symmetrical responses in mean change in Hadley Walker circulations between these two climates (related for the Hadley circulation to a dynamical or cooling effect induced by the ice-sheet )?

Other comments

- Please, provide error bars on the different bar plots

- Table 1 mentions past1000 simulations, but they are not used in the text.

- Make sure the color scales are identical for all the plots with the differences. Some of the values are so small that they should not be plotted. Would there be an interest to also show separately the results for models for which the difference is an increase in the index and the models for which it is a decrease? which would require that statistical significance is defined to tell for which models it is different from 0.

- For the maps of differences you could add isolines showing the pattern for PI to better highlight where the changes are located compared to the reference.

- Section 2 should be more informative. Details would be welcome to make sure we understand well how exactly the anomalies are computed for each of the periods, how the regressions are computed to provide the ensemble mean map, and also for each

model what is the level of significance for the regression and should nonsignificant values excluded (or set to 0) when computing the ensemble mean map? The estimates of the changes in variability are done using the ensemble mean value. Since the sampling is limited given the size of the model ensemble would it make any difference to consider the median valued?

- In section 3 tell why the observations look so noisy in figure 2.

- In section 3.1 PIcontrol should also be considered with historical to show the differences between this two close periods and discuss the limited length of the simulations. Some of the Pi Control experiments are long enough to be subsampled for an uncertainty analysis.

- Make sure the modes are discussed in the same order in all sections and figures.

- Even though the modes are extracted using an index and not EOF you could compute and provide the percentage of variance they represent. Previous studies Jolly et al. 2007 or Zhao et al. 2008 suggested that ENSO dominate variability in most models and thereby the teleconnection with the African monsoon, which is not the case in the observations. Is it valid here?

- P12 l230. The sentence is incorrect. Pausata et al. 2017 didn't simulate vegetation better they impose a mid-Holocene extreme reconstruction of the vegetation cover. So it should read something like when imposing mid Holocene vegetation reconstruction as boundary condition to the model.

Suggested references

- Joly, M., Voldoire, A., Douville, H., Terray, P., and Royer, J. F.: African monsoon teleconnections with tropical SSTs: validation and evolution in a set of IPCC4 simulations, Climate Dynamics, 29, 1-20, 2007.

- Zhao, Y., Braconnot, P., Harrison, S. P., Yiou, P., and Marti, O.: Simulated changes in the relationship between tropical ocean temperatures and the western African monsoon during the mid-Holocene, Climate Dynamics, 28, 533-551, 2007.

---

## Author Comment (AC1) · 3 Apr 2018

article

**1 Editor Comments**

Dear Author,

The reviews process of your manuscript it taking a long time, which is due to recurrent difficulties to get a second review. I am very sorry about it. Since time is running I propose that you go for a major revision of your manuscript, taking into account the important comments of Referee 1 who raised important issues on the clarity and organization of the paper, including methodological questions. I also include below my own expertise of the manuscript, focusing on major aspects. If you decide to provide a revised version of the manuscript it will be sent for a second round of reviews, with the hope that we do not have to face a long delay similar to the one of the first round.

Best regards

Pascale Braconnot

**Thank you for considering the length of time so far needed to find sufficient reviewers. We appreciate the opportunity to revise our manuscript in light of the comments so far from both yourself and Referee 1. We have outline how we would address these comments in the bold in the document below.**

1.1   Comments on the manuscript.

The subject is quite ambitious and timely, and the methodology used to discuss the two major Atlantic modes seems appropriate. As far as I know this has not been done yet, and providing systematic diagnoses to assess how the modes of variability are affected by climate change is a valuable task. In its present form however the paper is too descriptive and key aspects on precipitation are lacking. In particular :

- The introduction and first section highlight the fact that AMM and Atl3 modes have fundamental impact on South American and African monsoon, but this linkage is not discussed any further when considering the different climates. This limits the interest of the manuscript and is a major concern.

**We shall revise the text with this insight in mind. The later comment about the variability explained would provide a sensible approach to incorporate the monsoon in the discussion.**

- The discussion on physical and dynamical mechanisms should be enlarged. This concerns both the anomalous circulations associate with the SST modes and the changes in these circulations associated with changes in mode patterns in the different climates

**We shall enhance this style of discussion in a revised version.**

- One of the difficulties with the analyses of paleoclimate simulations is that both the background climate mean state and the variability change. How the pattern of the changes in variability is connected with patterns of the changes in the mean state should be discussed in more depth. A question out of this is does mode patterns only follow the mean state patterns? In other words if there is shift in the mode pattern is it directly reflecting a shift in the mean state pattern or is there other feedback that could explain that new areas become affected by the mode?

**We had attempted to answer these questions with the discussion section and Figs 12 & 13. Clearly these are insufficient (both in light of your comment and the lack of relationships demonstrated within them). We shall incorporate more individual analysis in our revised manuscript.**

- The outline of the paper is also a little bit "boring". This feeling is due to the fact that the discussion section could include additional analyses to explain when possible part of the rationale behind model responses (which may be different from one period to the other). The discussion section could thus be enlarged and have a more appealing title and content. It could compare relationships as it is done as well as mechanisms.

**We were already worried that the manuscript was too long. But we certainly agree that it is largely descriptive at present, and that further discussion of the mechanisms would increase its interest to readers (and reviewers). We shall endeavour to find some more insights to discuss in the revised manuscript.**

A few questions when reading the manuscript:

- It is interesting to see that the AMM mode is reduced at mid Holocene. Is it because the seasonal cycle is stronger and that a dipole-like pattern emerges in summer when comparing mid-Holocene with PI?

**That was our suspicion, but we shall investigate further for the revised manuscript.**

- Is there a reason why a colder climate would have increased variability?

**Recent work by Rehfeld et al (Nature, 2018) propose this would arise from increased temperature gradients. We shall refer to this and others in future.**

- Could the non-symmetrical differences between LGM and future results from non-symmetrical responses in mean change in Hadley & Walker circulations between these two climates (related for the Hadley circulation to a dynamical or cooling effect induced by the ice-sheet )?

**We are not sure why the non-symmetrical pattern occurs - clearly not because of the SST gradient changes proposed by previous authors as we investigated. The suggestion of changes in the Hadley and Walker cells is an interesting one, which we will look into.**

**1.2  Other comments**

- Please, provide error bars on the different bar plots

**We shall investigate whether there is a sensible method to compute the error bars. For some models they can be computed from the preindustrial, but not all.**

- Table 1 mentions past1000 simulations, but they are not used in the text.

**We have computed the AMM and ATL3 modes from these last millennium simulations (should someone want them). However, you are correct that they do not feature in the manuscript and should have been removed from the table.**

- Make sure the color scales are identical for all the plots with the differences. Some of the values are so small that they should not be plotted. Would there be an interest to also show separately the results for models for which the difference is an increase in the index and the models for which it is a decrease? which would require that statistical significance is defined to tell for which models it is different from 0.

**We honestly are not sure if there is a value in showing the individual model results in the manuscript. We had intended to provide all the model fields along with some code to visualise them as supplementary information. Nonetheless, we shall investigate these figures.**

- For the maps of differences you could add isolines showing the pattern for PI to better highlight where the changes are located compared to the reference.

**That is a good suggestion. We will have a go at doing this and see if it enhances the readability of the figures.**

- Section 2 should be more informative. Details would be welcome to make sure we understand well how exactly the anomalies are computed for each of the periods, how the regressions are computed to provide the ensemble mean map, and also for each model what is the level of significance for the regression and should non-significant values excluded (or set to 0) when computing the ensemble mean map?

**We shall explain our methods in more detail in a revised manuscript, as our current description is clearly insufficient. No significance testing was performed for the regression analysis - we shall investigate the practicalities of undertaking this during the revision stage.**

- The estimates of the changes in variability are done using the ensemble mean value. Since the sampling is limited given the size of the model ensemble would it make any difference to consider the median valued?

- In section 3 tell why the observations look so noisy in figure 2.

**That is a good question, and one that we were not sure about. We suspect they may be spectral echoes from the Reanalysis model, but shall investigate further.**

- In section 3.1 PIcontrol should also be considered with historical to show the differences between this two close periods and discuss the limited length of the simulations.

**We were worried that we were already showing too many panels and figures. However as both the Editor and Referee felt this was an important omission, we shall correct it.**

- Some of the Pi Control experiments are long enough to be subsampled for an uncertainty analysis.

**That is a good point. We will think about how to use them, and if they would be helpful (esp. given Referee 1's point about consistency between the models)**

- Make sure the modes are discussed in the same order in all sections and figures.

**This structure had not been rigidly observed in an attempt to keep a narrative flow. As it obviously did not help, we will revert to something more strict in future**

- Even though the modes are extracted using an index and not EOF you could compute and provide the percentage of variance they represent. Previous studies Jolly et al. 2007 or Zhao et al. 2008 suggested that ENSO dominate variability in most models and thereby the teleconnection with the African monsoon, which is not the case in the observations. Is it valid here?

**This is a really helpful suggestion of how to tie the SST variability back to the South American and African monsoons. We shall undertake this analysis in the revised manuscript.**

- P12 l230. The sentence is incorrect. Pausata et al. 2017 didn't simulate vegetation better they impose a mid-Holocene extreme reconstruction of the vegetation cover. So it should read something like when imposing mid Holocene vegetation reconstruction as boundary condition to the model.

**This was sloppy language on our behalf and we will edit as you suggest.**

**2   Referee 1**

This work investigated the change in the leading modes of the Tropical Atlantic Variability, the Atlantic Meridional Mode (AMM) and the Atlantic Niño (ATL3), in different

climate scenarios: the historical, the last glacial maximum, the mid-holocene and future simulations in the multi-model ensemble of the PMIP3/CMIP5. Authors used this set of experiments in order to find robust signal of change in the Tropical Atlantic Variability. They found that all models across all experiments are able to represent main characteristics of dominant modes of variability in the Tropical Atlantic in spite of the mean state bias.

The paper addressed a relevant question: how the Tropical Atlantic Variability change under different climates and how the information from mean state and past climates can be used as a constraint for the future. They quantified first the mean state model bias of the tropical temperature and precipitation in the historical simulations against reanalyses. After that, they compared the magnitude of the simulated change of the tropical temperature and precipitation in the mid-holocene, the lgm and the future climate with the mean state bias, concluding that the simulated changes are reasonably represented in these experiments. Hence, the main conclusion is that ATL3 and AMM are well represented among models and experiments considered, although authors found weak correlation with change in temperature gradients, so it is not possible to identify emerging constraints for future projections from this analysis.

- I think this is a good work but sometimes the storyline is hard to follow: I suggest to work more on the structure of the paper and on discussion and conclusions in order to clarify main findings (perhaps merging both sections would be helpful).

**This suggestion is echoed by the editor. We shall take this onboard during our revisions.**

- About the method, I think that conclusions hold only if same models among experiments are considered, otherwise results might be affected by different model physics and also by the different number of the model used for each experiment (see my comments below).

**We feel that this is overly conservative, but propose to test the suggestion in a revised manuscript.**

- Furthermore, several typos in the captions (see specific comments below) made the paper difficult to read.

- Some recent and important literature is also missing (see the list below).

**In a revised manuscript we shall engage better with the literature about the ITCZ, especially in its formal sense.**

2.1   Minor Comments

There are some typos in the text. Here, it is a list:

- Ln 39: "It is associated with a shift ...". It is not clear what is the subject of the sentence.

- Ln 63: a full stop is missing after the brackets "(c.f. the AMM)".

- Figure 1: the caption mentioned HadISST for panels a and b, but 20C_reanalysis is written on the top of both panels.

- Figure 2: replace "in precipitation" with "of the precipitation" in the caption.

- Ln 203: You specified already the acronym TAV for Tropical Atlantic Variability in the very beginning of the paper. You can use it throughout the paper.

- Figure 3: Typos in the caption. ATL3 is shown in panel a and c. AMM in panel b and d. In the upper panels observation results and in the lower panels simulation results are shown. Also the standard deviations must be reorder: 0.17 refers to panel a, 0.18 to panel b, 0.05 to panel b and 0.04 to panel d.

- Figure 6, 8, 10: It is better to change the color of dark gray bars in light gray bars. You could also replace the letters associated to each model, with a number or a short name.

- Ln 280: repetition: "... is expected to be still be..."

**We apologise that so many typographical errors passed through our proofreading. We shall correct all of them in a revised version.**

**3   Major Comments**

- Ln 36: "...strength of the ITCZ". What do you mean for strength of the ITCZ. Usually, the ITCZ is the latitude of the wind convergence, and or, the latitude of the maximum of the precipitation. Please clarify.

- Ln 40-42: Add literature about the ITCZ. (e.g. Schneider et al., 2014, Bischoff, T., & Schneider, T. , 2016, Green, B., & Marshall, J., 2017).

- Ln 63: cite Schneider et al., 2014.

- Ln 68: You might want to cite also D'Agostino et al., 2017. They linked Hadley Circulation changes also to change of the meridional temperature gradient and interhemispheric thermal contrast.

- Ln 166-168: What do you mean for: "models are unable to get the full intensity of the ITCZ"? I would like to clarify again that the ITCZ is a latitude of the maximum of the tropical precipitation not the rainfall intensity itself. Please reword, otherwise add specific analysis on the ITCZ shift, including also how to define the ITCZ in the methods.

[Figure]

- Ln 335-340: these statements about the ITCZ are not supported by the analysis shown in the paper. How did you quantify the ITCZ shift? Please include further analysis

**We confess that we had used ITCZ as shorthand for rain within the zone, rather than its formal definition of as a location. So, for example, by the text in lines 166-168 we mean there was insufficient rain falling within the ITCZ. We apologise for this sloppiness and will correct in a revised version. We had not undertaken a formal analysis of the ITCZ location. We shall look into the logistics of performing such an analysis for the revised manuscript.**

- Ln 225, the Mid-Holocene section: the Tropical Atlantic is elsewhere cooler than Pre-Industrial and this is pretty consistent among models. I was wondering if you believe to this result if the magnitude of the change is weaker than the change in the mean state.

**We suspect this result is robust, but have not yet thought of a convincing mechanism to explain it. We shall comment on this in the revised manuscript.**

- Figure 4: I have a question about these panels. How did you perform the difference between the multimodel ensemble mean for each experiment and the multimodel ensemble mean of the Pre-industrial condition? You must use different pre-industrial multimodel ensemble mean for each experiments, because you must account for different model list. I did not find any specification about it in the paper.

**We apologise for the lack of precise explanation. We have averaged the *changes* from each model - as we thought was standard approach. We shall revise the methodology to be much more explicit about this**

- Furthermore, I have some doubt: I don't think the mean state bias can be used to give credibility to results of the different experiments in this context, because the ensemble mean of the historical experiment accounts for 14 models. The ensemble mean of the pre-industrial for 21. They tell different story then. The difference is much more evident if you compute different pre-industrial ensemble mean for each experiment! Therefore, when you compare the climate change with the mean state bias, you are wrong because the different model list. I suggest to restrict the model list to common models only for all experiments. Unfortunately, the list is very short (only 9 models) but I think that is still possible to reach robust conclusions.

**We understand your point. We had assumed the biases to not vary much between the historical and preindustrial experiments and the various ensembles. We shall repeat the analysis with the subset of model consistent across all experiments to check this assumption for the revised manuscript.**

- Ln 235: add also dust and cite Erger et al., 2016.

- Ln 316: please quantify a "little relationship".

- Ln 330: please quantify again "little robust relationship".

**By 'little', we meant neither visually nor statistically significant. We shall rephrase We understand your point. We had assumed the biases to vary much between the historical and preindustrial experiments and the various ensembles. It is easy enough to repeat the all the analysis with the subset of model consistent across all experiments to check this assumption.**

---

## Author Response (AR1)

**1 Overall Response**

We would like to thank both Pascale (as the Editor) and the reviewer for their comments on our work. We believe that we have taken them onboard and hopefully the manuscript is more appealing for other reviewers and readers to look at now. We itemize below our thoughts and any edits made in response to the individual comments.

The major changes implemented for this revised version of the manuscript are:

- We have adopted a different definition of the AMM. This was motivated, so that the manuscript conforms with the most recent release of the CVDP software.
- Relationship of modes with rainfall brought earlier in the text when the patterns related to the AMM and ATL3 modes are compared to observations/reanalysis (added new Figure 4).
- Reference to other basins substantially down-weighted in the text, focus only on the Atlantic ocean region.
- Figures were redone:
  - 1. adjusted to show just the focus of the paper, the Atlantic ocean region.
  - 2. preindustrial control results were added to Figures 1,2 and 3
  - 3. new Figure 4 with historical rainfall patterns associated with the ATL3 and AMM modes.
  - 4. Figures 6,8 and 10 have changes in the precipitation response as well as contour lines showing PI-control patterns.
- Results were recalculated in order to consider just the subset of models that submitted consistent results across all experiments.

**2 Editor Comments on the manuscript.**

The subject is quite ambitious and timely, and the methodology used to discuss the two major Atlantic modes seems appropriate. As far as I know this has not been done yet, and providing systematic diagnoses to assess how the modes of variability are affected by climate change is a valuable task. In its present form however the paper is too descriptive and key aspects on precipitation are lacking. In particular :

• The introduction and first section highlight the fact that AMM and Atl3 modes have fundamental impact on South American and African monsoon, but this linkage is not discussed any further when considering the different climates. This limits the interest of the manuscript and is a major concern.

We have increased the discussion of the precipitation consequences dramatically and brought this earlier in the text. This is most visible through the inclusion of precipitation patterns in many of the figures. We hope that by stressing that the tropical modes of variability define the SST anomalies that ultimately control ITCZ-related precipitation

• The discussion on physical and dynamical mechanisms should be enlarged. This concerns both the anomalous circulations associated with the SST modes and the changes in these circulations associated with changes in mode patterns in the different climates

**We have attempted to increase discussion around the physical and dynamical mechanisms. We have created a new section of the manuscript specifically focused on this topic.**

• One of the difficulties with the analyses of paleoclimate simulations is that both the background climate mean state and the variability change. How the pattern of the changes in variability is connected with patterns of the changes in the mean state should be discussed in more depth. A question out of this is does mode patterns only follow the mean state patterns? In other words if there is shift in the mode pattern is it directly reflecting a shift in the mean state pattern or is there other feedback that could explain that new areas become affected by the mode?

The spatial patterns associated with each of the Tropical modes are very robust and closely related to the SST anomalies. Mode shifts actually reflect changes in intensity/amplitude rather than changes in spatial distribution. Figures 13 and 14 quantify the strength of the modes as a function of the SST distribution showing that for the AMM there is clear distinction between the climatic periods. The LGM and Mid-Holocene show opposite behavior. In the LGM there is an increase in the AMM amplitude while the North-South SST gradient decreases while for the Mid-Holocene there is an increase in the North-South SST gradient accompanied by a decrease in the amplitude of the AMM. The behavior of the AMM for the 1pctCO2 shows an overall weakening of the AMM mode with a decrease in both the AMM amplitude and the associated North-South SST gradient. For the ATL3, the relationship between the amplitude and east-West SST gradient for each climatic period is not as clear cut. In other words, changes in the variability of the AMM mode appear directly related to changes in the mean state. However for the ATL3, the relationship is not that linear

• The outline of the paper is also a little bit boring. This feeling is due to the fact that the discussion section could include additional analyses to explain

when possible part of the rationale behind model responses (which may be different from one period to the other). The discussion section could thus be enlarged and have a more appealing title and content. It could compare relationships as it is done as well as mechanisms.

We have introduced another section "TAV amplitude changes as a function of the SST gradient" for the relationship between changes in the AMM and ATL3 relative to changes in the meridional and zonal gradients, respectively. Furthermore, Discussions and Conclusion section were merged

A few questions when reading the manuscript:

• It is interesting to see that the AMM mode is reduced at mid Holocene. Is it because the seasonal cycle is stronger and that a dipole-like pattern emerges in summer when comparing mid-Holocene with PI?

We believe that the AMM mode is reduced because the MH is cooler in the Tropical Atlantic compared to PI, even though the seasonal cycle is stronger.

• Is there a reason why a colder climate would have increased variability?

We do not think a priori that cooling should lead to increased variability. The IPCC suggests that monsoon region variability should increase with temperature (Christensen et al., 2013). Yet Rehfeld et al. (2018) have just shown (similar to our findings here) that variability was substantially higher at the LGM. They suggest this is because of stronger meridional temperature gradients, but we don't find particular evidence for that for the AMM.

• Could the non-symmetrical differences between LGM and future results from non-symmetrical responses in mean change in Hadley & Walker circulations between these two climates (related for the Hadley circulation to a dynamical or cooling effect induced by the ice-sheet)?

Yes, they could. The impact of different climates on the Hadley Circulation in the Tropical Atlantic is subject of ongoing research. For example, Jones et al. (2018) show that the Walker circulation responds to thresholds in ice-sheet volume during the deglaciation - causing remote changes in ENSO teleconnections.

**2.1 Other comments**

• Please, provide error bars on the different bar plots

We investigated the practicality of providing error bars on these plots. For some models, they could have been readily computed by subsetting the long preindustrial control simulations, but this cannot be done for all models. We have therefore decided to leave off all the models to maintain consistency.

• Table 1 mentions past1000 simulations, but they are not used in the text.

**Table 1 has been modified to reflect only the simulations used in this study**

• Make sure the color scales are identical for all the plots with the differences. Some of the values are so small that they should not be plotted.

**The color scales have been standardised and small changes removed.**

• Would there be an interest to also show separately the results for models for which the difference is an increase in the index and the models for which it is a decrease? which would require that statistical significance is defined to tell for which models it is different from 0.

We feel that the manuscript is already long enough. So we would prefer not to include additional figures showing this decomposition. Figures 13 and 14 show the individual standard deviations for the AMM and ATL3 modes for each model. The patterns for all individual models are available online.

• For the maps of differences you could add isolines showing the pattern for PI to better highlight where the changes are located compared to the reference.

**Isolines have now been incorporated into every plot where relevant.**

• Section 2 should be more informative. Details would be welcome to make sure we understand well how exactly the anomalies are computed for each of the periods, how the regressions are computed to provide the ensemble mean map, and also for each model what is the level of significance for the regression and should non-significant values excluded (or set to 0) when computing the ensemble mean map?

**Efforts were made to better detail the methodology employed.**

• The estimates of the changes in variability are done using the ensemble mean value. Since the sampling is limited given the size of the model ensemble would it make any difference to consider the median valued?

We feel that explicitly providing a mean is the standard practice the changes for each ensemble member are shown in the bar-charts in both absolute and percentage terms, so the full distribution is there for readers who wish to see that.

• In section 3 tell why the observations look so noisy in figure 2.

This is a feature of the  $20^{th}$  Century reanalysis precipitation. It likely occurs as a spectral echo from the sharp topography in the region. We have decided to shift to using the GPCP rainfall product for this particular plot. The subsequent plots still use the  $20^{th}$  Century reanalysis, because we feel that the long, consistent nature of this dataset should be better at capturing variability.

• In section 3.1 PIcontrol should also be considered with historical to show the differences between this two close periods and discuss the limited length of the simulations.

PI-control panels were added to Figures 1-3 in section 3. We had previously assumed that including these would be detrimental to the length of the manuscript. As both reviews request this information, we were wrong in this assumption.

• Some of the Pi Control experiments are long enough to be subsampled for an uncertainty analysis.

Uncertainty analysis for individual PI-control simulations for each of the models is interesting in itself, but we believe it is not essential for the understanding of the ensemble mean changes approached here.

• Make sure the modes are discussed in the same order in all sections and figures.

In the revised manuscript, ATL3 is discussed first followed by the AMM

• Even though the modes are extracted using an index and not EOF you could compute and provide the percentage of variance they represent. Previous studies Jolly et al. 2007 or Zhao et al. 2008 suggested that ENSO dominate variability in most models and thereby the teleconnection with the African monsoon, which is not the case in the observations. Is it valid here?

We have decided to focus the study just on the tropical Atlantic modes of variability excluding teleconections from the Pacific (i.e. ENSO). A new Figure (Figure 4) was added to show the relationship of the ATL3 and AMM modes with PPT. This is done through the regression of the respective indices onto the PPT fields. We choose to show these as absolute rainfall variations, rather than percentages, for consistency. However, we acknowledge that this approach can over-emphasize their importance.

• P12 l230. The sentence is incorrect. Pausata et al. 2017 didnt simulate vegetation better they impose a mid-Holocene extreme reconstruction of the vegetation cover. So it should read something like when imposing mid Holocene vegetation reconstruction as boundary condition to the model.

This sentence was changed to "It has been shown that when imposing mid Holocene vegetation reconstruction as boundary condition to the model, inter-annual climate variability can be impacted".

**3 Referee 1**

This work investigated the change in the leading modes of the Tropical Atlantic Variability, the Atlantic Meridional Mode (AMM) and the Atlantic Nio (ATL3), in different climate scenarios: the historical, the last glacial maximum, the mid-holocene and future simulations in the multi-model ensemble of the PMIP3/CMIP5. Authors used this set of experiments in order to find robust signal of change in the Tropical Atlantic Variability. They found that all models across all experiments are able to represent main characteristics of dominant modes of variability in the Tropical Atlantic in spite of the mean state bias.

The paper addressed a relevant question: how the Tropical Atlantic Variability change under different climates and how the information from mean state and past climates can be used as a constraint for the future. They quantified first the mean state model bias of the tropical temperature and precipitation in the historical simulations against reanalyses. After that, they compared the magnitude of the simulated change of the tropical temperature and precipitation in the mid-holocene, the lgm and the future climate with the mean state bias, concluding that the simulated changes are reasonably represented in these experiments. Hence, the main conclusion is that ATL3 and AMM are well represented among models and experiments considered, although authors found weak correlation with change in temperature gradients, so it is not possible to identify emerging constraints for future projections from this analysis.

• I think this is a good work but sometimes the storyline is hard to follow: I suggest to work more on the structure of the paper and on discussion and conclusions in order to clarify main findings (perhaps merging both sections would be helpful). This suggestion is echoed by the editor. We added a new section and "Discussions" and "Conclusions" were merged.

• About the method, I think that conclusions hold only if same models among experiments are considered, otherwise results might be affected by different model physics and also by the different number of the model used for each experiment (see my comments below).

We believe this is an overly conservative position. Nonetheless, results have been recomputed with just the subset of models that submitted simulations for all climate periods. These subset figures are appended at the end of this response for your perusal.

• Furthermore, several typos in the captions (see specific comments below) made the paper difficult to read.

We have addressed all typos

• Some recent and important literature is also missing (see the list below).

We have included all references suggested except Egerer et al. (2016) on dust because the sentence was rephrased (following comment fom Reviewer 2 that better vegetation was not a result of simulation but of boundary condition used)

**3.1** Minor Comments**

There are some typos in the text. Here, it is a list:

• Ln 39: It is associated with a shift ...". It is not clear what is the subject of the sentence.

the sentence was re-written as "The AMM is associated with a shift ..."

• Ln 63: a full stop is missing after the brackets "(c.f. the AMM).

full stop was added

• Figure 1: the caption mentioned HadISST for panels a and b, but 20C\_reanalysis is written on the top of both panels.

**Labels on figures corrected**

• Figure 2: replace in precipitation with of the precipitation in the caption.

**replaced**

• Ln 203: You specified already the acronym TAV for Tropical Atlantic Variability in the very beginning of the paper. You can use it throughout the paper.

**We have now used the acronym TAV instead of "Tropical Atlantic Variability" when found throughout the text.**

• Figure 3: Typos in the caption. ATL3 is shown in panel a and c. AMM in panel b and d. In the upper panels observation results and in the lower panels simulation results are shown. Also the standard deviations must be reorder: 0.17 refers to panel a, 0.18 to panel b, 0.05 to panel b and 0.04 to panel d.

**The labels on the figures have now been corrected.**

• Figure 6, 8, 10: It is better to change the color of dark gray bars in light gray bars. You could also replace the letters associated to each model, with a number or a short name.

**We have changed the color from dark gray to light gray**

• Ln 280: repetition: ... is expected to be still be..."

**corrected the sentence**

**4 Major Comments**

• Ln 36: ...strength of the ITCZ. What do you mean for strength of the ITCZ. Usually, the ITCZ is the latitude of the wind convergence, and or, the latitude of the maximum of the precipitation. Please clarify.

**We have revised in the text to reflect the intensity of the ITCZ-related PPT.**

• Ln 40-42: Add literature about the ITCZ. (e.g. Schneider et al., 2014, Bischoff, T., & Schneider, T., 2016, Green, B., & Marshall, J., 2017).

**Discussion of this relevant literature on the ITCZ was added.**

• Ln 63: cite Schneider et al., 2014.

**This review paper on the migration/dynamics of ITCZ is cited.**

• Ln 68: You might want to cite also DAgostino et al., 2017. They linked Hadley Circulation changes also to change of the meridional temperature gradient and interhemispheric thermal contrast.

**D'Agostino et al., 2017 is cited**

• Ln 166-168: What do you mean for: models are unable to get the full intensity of the ITCZ? I would like to clarify again that the ITCZ is a latitude of the maximum of the tropical precipitation not the rainfall intensity itself. Please reword, otherwise add specific analysis on the ITCZ shift, including also how to define the ITCZ in the methods.

**The reviewer is correct. The phrase was reworded. What was meant is that models are unable to realistically represent the distribution of the ITCZ-related rainfall.**

• Ln 335-340: these statements about the ITCZ are not supported by the analysis shown in the paper. How did you quantify the ITCZ shift? Please include further analysis

We agree that the ITCZ position shift was not quantified, and was sloppy language on our behalf. The change or displacement of rainfall mentioned is in reference to the Tropical Atlantic ITCZ-related precipitation, this was changed in the text.

• Ln 225, the Mid-Holocene section: the Tropical Atlantic is elsewhere cooler than Pre-Industrial and this is pretty consistent among models. I was wondering if you believe to this result if the magnitude of the change is weaker than the change in the mean state.

**Yes, this cooling is also noted in other studies that evaluate PMIP data such as Braconnot et al. (2007) and Zhao et al. (2005).**

• Figure 4: I have a question about these panels. How did you perform the difference between the multimodel ensemble mean for each experiment and the multimodel ensemble mean of the Pre-industrial condition? You must use different pre-industrial multimodel ensemble mean for each experiments, because you must account for different model list. I did not find any specification about it in the paper.

We have now increased the explanation of the methodology. We had performed the average of the change in pattern (rather than the change in the ensemble mean patterns). This is the ensemble averaging approach adopted by the IPCC. Whilst this does mean that there is some differences in the pre-industrial multimodel means there are relatively small (you can see their impact through the new isolines added on the figures).

• Furthermore, I have some doubt: I dont think the mean state bias can be used to give credibility to results of the different experiments in this context, because the ensemble mean of the historical experiment accounts for 14 models. The ensemble mean of the pre-industrial for 21. They tell different story then. The difference is much more evident if you compute different pre-industrial ensemble mean for each experiment! Therefore, when you compare the climate change with the mean state bias, you are wrong because the different model list. I suggest to restrict the model list to common models only for all experiments. Unfortunately, the list is very short (only 9 models) but I think that is still possible to reach robust conclusions.

We have added the preindustrial control ensemble mean to Figs 1 to 4. The analysis has been repeated with just the subset of models consistent across all experiments. The versions of these figures are included at the end of this response.

• Ln 235: add also dust and cite Erger et al., 2016.

This has not included, because the sentence was rephrased following comment from Reviewer 2.

• Ln 316: please quantify a little relationship.

"little relationship" was substituted for "no apparent relationship emerging"

• Ln 330: please quantify again little robust relationship. The text was re-written as: "A decrease in the AMM amplitude change can be associated with an increase in the meridional SST gradient for the mid-Holocene while the opposite occurs for the 1% per year until quadrupled  $CO_2$  (1pctCO2). The relationship at LGM is more dubious relative to the changes in SST-gradient but suggests an increase in the AMM amplitude. Overall, the relationship between the amplitude change of the AMM and the changes in the meridional SST-gradient depends on the climatic period considered."

**References**

- Braconnot, P., Otto-Bliesner, B., Harrison, S., Joussaume, S., Peterchmitt, J.-Y., Abe-Ouchi, A., Crucifix, M., Driesschaert, E., Fichefet, T., Hewitt, C., et al.: Results of PMIP2 coupled simulations of the Mid-Holocene and Last Glacial Maximum–Part 2: feedbacks with emphasis on the location of the ITCZ and mid-and high latitudes heat budget, Climate of the Past, 3, 279–296, 2007.
- Christensen et al.: Climate phenomena and their relevance for future regional climate change [IPCC WG1 AR5 Chap14], 2013.
- Jones, T. R., Roberts, W. H., Steig, E. J., Cuffey, K., Markle, B., and White, J.: Southern Hemisphere climate variability forced by Northern Hemisphere ice-sheet topography, Nature, 554, 351, 2018.
- Rehfeld, K., Münch, T., Ho, S. L., and Laepple, T.: Global patterns of declining temperature variability from the Last Glacial Maximum to the Holocene, Nature, 554, 356, 2018.
- Zhao, Y., Braconnot, P., Marti, O., Harrison, S., Hewitt, C., Kitoh, A., Liu, Z., Mikolajewicz, U., Otto-Bliesner, B., and Weber, S.: A multi-model analysis of the role of the ocean on the African and Indian monsoon during the mid-Holocene, Climate Dynamics, 25, 777–800, 2005.

---

## Author Response (AR2)

**1 Overall Response**

**We thank the editor and referees for the timely and reasonable reviews of our revised manuscript. Below we respond to the individual comments. We appreciate the diligence and kind words said about our research.**

**2 Review #1**

- This work investigated the change in the leading modes of the Tropical Atlantic Variability the Atlantic Meridional Mode (AMM) and the Atlantico Nio (ATL3) in different climate scenarios: the historical, the last glacial maximum, the mid-holocene and future simulations in the multi-model ensemble of the PMIP3/CMIP5. Authors used this set of experiments in order to find robust signal of change in the Tropical Atlantic Variability. They found that all models across all experiments are able to represent main characteristics of dominant modes of variability in the Tropical Atlantic in spite of the mean state bias.

- After the revision, the paper is improved and authors addressed carefully referee comments on the previous version of the manuscript. I am fine with that. More work is needed on some typos or considering to rephrase some sentence. I therefore I recommend a minor revision following comments below and I suggest to publish the paper in a close version to the current one.

**We are glad that the reviewer found our revised manuscript a better version. We thank them for their diligent suggestions that lead to that improvement.**

**2.1 Minor Comments**

- There are some typos in the figures: the i somehow appears as an l. Please correct.

**We can assure the reviewer that these are "i" in the plotting scripts. This relates to the rendering of the PDF images and we will discuss this with the Technical Editors.**

- Pg. 2, Ln 29: Typo: to follows remove the s.

**Corrected**

- Pg. 2 Ln 30-31: The jump in the flow is to abrupt. Consider rephrasing. This sentence Schneider et al. (2014) presents a comprehensive review of the dynamics of the ITCZ and associated meridional shifts seems to be out of the contest.

The pair of sentences have been rephrased to "Put simply, the ITCZ tends to follow the warmest hemisphere - although see Schneider et al. for a description of true nuances of this relationship."

- Pg. 3 Ln 5: A blank is missing after the full stop.

- Pg. 3 Ln 19: positioning -¿ position

**Corrected**

- Pg. 4 Ln 2: I would mention the last millennium here since your are not using it in the study.

**We feel that if we do not mention the Last Millennium simulations here that we are underselling the effort of PMIP. We therefore would like to keep this sentence in the manuscript.**

- Pg. 4 Ln 5: The climatology  is used -¿ is calculated.

**Corrected**

- Pg .6 Ln 27:  The Atlantic is warmest on the Equator is the warmest . ? I do not understand Consider to rephrase.

**We mean that the highest SSTs in the Atlantic occur on the Equator. We have rephrased this sentence, but feel the sentence was correct in its original phrasing.**

- Pg. 7 Ln 8-9: they have too much rain falling south of the ITCZ -¿  they show too strong positive precipitation anomaly on the southward flank of the ITCZ. Please be more accurate in the description of Figure and use proper English.

**This has been rephrased.**

- Pg. 7 Ln 9:  whilst the models simulate too much convection over N.E. Brazil -¿ rephrase. You are not showing specifically convective precipitation only, but instead the total precipitation. I fully understand that the monsoonal rainfall is mainly due to the convection, but it is inappropriate to use the term convection here.

**We have rephrased this sentence. We have kept the mechanistic reference to convection, but stressed that we have not proved this mechanism with our analysis.**

- Pg. 8 Ln 18: typo: onto not on to.

- Pg. 8 Ln 29:  regressions with from the reanalysis. Remove with.

**Both corrected**

- Pg. 8 Ln 30:  PPT patterns  what is PPT pattern?

**PPT is shorthand for precipitation. It has been replaced by "rainfall"**

- Pg. 11 Ln 21: typo timeseries -¿ time series.

**corrected**

**2.2   Major Comments**

- Pg. 2 Ln 1: the latitudinal displacement of the rain producing Inter-Tropical Convergence Zone. I think that the rain belt is produced by the convergence of the surface winds into the ITCZ where air masses converge and uplift inside the Hadley Cell. This is where the rain comes. Consider to rephrase.

**With this sentence we are aiming for a general introduction to tropical meteorology. We feel that this first paragraph is not the place to get into semantics about ITCZ definitions.**

- Pg. 2 Ln 15: most well-known mode of TAV is you did not specified TAV acronym before in the Introduction. Correct please.

**TAV is defined in the abstract. We have now repeated the definition again here.**

- Pg. 3 Ln 22: The leading mode is a meridional mode. The leading mode of what? It is not clear in the text.

**We have inserted "of TAV"**

- Pg. 3 Ln 24: The inter hemispheric SST-gradient is accompanied by a cross-equatorial atmospheric flow in the same direction. I do not understand what you mean. I think that this sentence is incorrect. Just to clarify the concept: if the northern hemisphere is warmer than the southern hemisphere (e.g. strong inter-hemispheric thermal contrast), then the cross-equatorial (winter - SH) Hadley Circulation is stronger than the summer counterpart. Therefore the Atmospheric Heat transport (AHT) is southward, following the cross-equatorial flow. From McGee et al., 2014: In the solsticial seasons, the (cross-equatorial) Hadley circulation transports 2.5 PW of heat into the winter hemisphere, and the magnitude of this seasonal AHTEQ is linearly related to how far the ITCZ located near the boundary between the Hadley cells  migrates into the summer hemisphere. Therefore, the more the ITCZ migrates into the summer hemisphere, stronger the AHT toward the winter hemisphere is. Continuing from McGee et al., 2014: the ITCZs position in the Northern Hemisphere and the associated southward AHTEQ is driven by the cross-equatorial ocean heat transport (OHTEQ), which has a magnitude of 0.4 PW and is principally driven by the Atlantic Oceans meridional overturning circulation (AMOC). Changes in ITCZ position, and thus in AHTEQ, may therefore provide insight into past changes in heat transport by the AMOC; alternatively, they may reflect changes in the hemispheric balance of TOA energy fluxes (e.g., asymmetries in hemispheric albedo). [..]  a southward shift of the ITCZ during Heinrich Stadial S1 would increase AHT into the NH, compensating for a reduction or shutdown of the AMOC. Consider to rephrase at Ln 24 in order to make it clear what you mean.

**We have removed a specification of the direction of the atmospheric flow. We want the reader to take away an impression that changes in the gradient should be related to changes in the atmospheric circulation.**

- Pg. 3 Ln 1-18: Maybe it is matter of personal taste, but I suggest to work a bit on this paragraph. It seems you are listing only facts. Try to rephrase.

**We understand your suggestion. However we want to stress that this section is a complete literature review. And that there is not much literature. We feel the dull "listing" style of prose conveys this most effectively.**

- Pg. 4 Ln 14  although only in the case of GFDLs last glacial maximum run was this for scientific rather than resourcing reasons. It is not clear to me this sentence. Consider rephrasing.

**This is meant to say that GFDL did not manage to complete the run because it cannot cope with the LGM conditions. We feel this point is worth making, but had failed to notice that CSIRO-Mk3L did not complete: https://wiki.lsce.ipsl.fr/pmip3/doku.php/pmip3:database:status**

- Pg. 8 Ln 18-19: This sentence seems out of context. Why are you talking about ENSO now?

**We had included this text to explain the blob of red in the Pacific in Fig. 8b. In the revised text, we have rephrased this and pointed to the new supplementary table.**

- Figure 8: Indeed, there are big differences between ATL3 and AMM with respect to Fig.1 in the response to reviewers recomputed using the common models for the PI ensemble mean. Can you comment on that? Furthermore, correct the colorbar: symmetric labels with respect to the zero.

**We have further discussed the caveat highlighted above by including an additional sentence stating "Again we must stress that the patterns shown are the ensemble mean and may average out some substantial variation in response between the individual models."**

- Figure 10 and 12: Correct the colorbar: symmetric labels with respect to the zero.

**We understand why this has been requested. However, given the units, the fontsize must be reduced dramatically to fit in labelling for every single interval. Instead we have included a note in the caption explicitly stating that the contours are linear - allowing the reader to determine the other labels.**

- Figure 13 and 14: Add a legend with colored dots associated to each experiment. Also Numbers associated to each model would be interesting to show, as well as showing the multimodel ensemble mean for each experiment.

**A legend has been added to explain the colors. We experimented with labeling each individual model, but found that it overcomplicated the image.**

**3 Review #2**

- This is a useful paper that examines the response of Atlantic climate variability (the Atlantic Nino mode and the Atlantic Meridional Mode) to climate change in the past (Holocene, LGM) and future. Although there is no significant scientific insight gained from the work, the paper presents the first systematic examination of Atlantic variability across the three sets of experiments and therefore provide a useful reference that can be used for future studies.

**We agree that this work is mainly descriptive - yet provides a solid initial assessment for other researchers to build upon.**

- Scientifically, perhaps, the most interesting point is that they have no conclusion what caused the change of these modes. In particular, there is no relation between E-W SST gradient and the amplitude of ALT3, and there is no relation between the N-S SST gradient and the AMM, although both relationships have been suggested as a mechanism in previous works. Although this is a negative conclusion, it is interesting in that sense that the result here questions our current understanding of the mechanism of these Atlantic modes. Indeed, the mechanism of Atlantic variability has been studied much less than their Pacific, and perhaps the Indian Ocean, counterparts. This study is a confirmation of this. I recommend the publication of the paper after addressing one major question below.

**We were rather surprised by the lack of these relationships. We feel that paleo-simulations should be used more regularly to assess the viability of mechanisms proposed to explain future changes. It is nice to see that the Referee shares this opinion.**

- It will be also interesting to show ENSO variability here for two purposes.

- First of all, it gives a contrast example in the Pacific. Second, and perhaps more important, ENSO exerts significant impact on tropical Atlantic and may therefore have some impact on the response of the variability too. If, for example, there is no relation between the change of ENSO strength and the Atlantic modes, it is also an interesting negative conclusion that the response of ENSO is not causing the change of the Atlantic modes.

- In this regard, it will be interesting to show the correlation among: ALT3, AMM and ENSO (perhaps NINO3.4), may be as a table, in the observation and the models.

**The initial submission of this manuscript had a fair bit of discussion about ENSO. That was motivated by the fact that the regression patterns were plotted for the whole globe and naturally showed a response in the Tropical Pacific. In response to the previous round of reviews, it was suggested that we only consider a subset of models. In testing this suggestion, we discovered that the composition of the ensemble substantially altered the ENSO relationships plotted. We therefore moved to regional-only plots for the revised manuscript seen by this reviewer. Creating a large table is much better way of showing this susceptibility. We have created this table, but it is so large that we are only including it as a supplemental. We highlight the existence of this table when ENSO is discussed in relation to Fig. 8b.**

[revised manuscript text omitted]

---

## Author Response (AR3)

**UCL DEPARTMENT OF GEOGRAPHY**

[Figure]

Pascale Braconnot
Editor
Climate of the Past

23 August 2018

Dear Pascale,

Thank you for your thoughts about Ilana and my manuscript.

I must apologise that the supplement was not provided. I was having large problems with the upload procedure on the website for the file and after trying again today are still having problems. It is currently available for inspection at http://www2.geog.ucl.ac.uk/~ucfaccb/BrierleyWainer2018.cp-2017-145.zip . I am emailing the editorial team to see if they can attach it manually to the manuscript records and press submit for me.

I also had not realised that the Nature Climate Change was seen as the official documentation for PMIP3. I've now referenced at the first introduction for PMIP. I have also added a citation to the Zhao et al. paper.

I hope that you find this sufficient information to initiate the publication procedure. However, do not hesitate to ask me if you would like something further.

Yours sincerely,
Chris Brierley

UCL Department of Geography
University College London, Pearson Building, Gower Street, London, WC1E 6BT
Tel:  +44 (0)20 7679 0571    Fax:  +44 (0)20 7679 7565
c.brierley@ucl.ac.uk
http://www.geog.ucl.ac.uk/about-the-department/people/academics/chris-brierley/